



# Meridional and vertical variations of the water vapour isotopic composition in the marine boundary layer over the Atlantic and Southern Ocean

Iris Thurnherr[1], Anna Kozachek[2], Pascal Graf[1], Yongbiao Weng[3,4], Dimitri Bolshiyanov[2], Sebastian Landwehr[5], Stephan Pfahl[1,6], Julia Schmale[5], Harald Sodemann[3,4], Hans Christian Steen-Larsen[3,4], Alessandro Toffoli[7], Heini Wernli[1], and Franziska Aemisegger[1]

[1]Institute for Atmospheric and Climate Science, ETH Zürich, Zurich, Switzerland
[2]Climate and Environmental Research Laboratory, Arctic and Antarctic Research Institute, St Petersburg, Russia
[3]Geophysical Institute, University of Bergen, Bergen, Norway
[4]Bjerknes Centre for Climate Research, Bergen, Norway
[5]Laboratory of Atmospheric Chemistry, Paul Scherrer Institute, Villigen PSI, Switzerland
[6]Institute of Meteorology, Freie Universität Berlin, Berlin, Germany
[7]Department of Infrastructure Engineering, The University of Melbourne, Melbourne, Australia

**Correspondence:** Iris Thurnherr (iris.thurnherr@env.ethz.ch)

**Abstract.** Stable water isotopologues (SWIs) are useful tracers of moist diabatic processes in the atmospheric water cycle. They provide a framework to analyse moist processes on a range of time scales from large-scale moisture transport to cloud formation, precipitation, and small-scale turbulent mixing. Laser spectrometric measurements on research vessels produce high-resolution time series of the variability of the water vapour isotopic composition in the marine boundary layer. In this

study, we present a five-month continuous time series of such ship-based measurements of $\delta^2$H and $\delta^{18}$O from the Antarctic Circumnavigation Expedition (ACE) in the Atlantic and the Southern Ocean in the time period from November 2016 to April 2017. We analyse the drivers of meridional SWI variations in the marine boundary layer across diverse climate zones in the Atlantic and Southern Ocean using Lagrangian moisture source diagnostics and relate vertical SWI differences to near-surface wind speed and ocean surface state. The median values of $\delta^{18}$O, $\delta^2$H and d-excess during ACE decrease continuously from

low to high latitudes. These meridional SWI distributions reflect climatic conditions at the measurement and moisture source sites, such as air temperature, specific humidity, and relative humidity with respect to sea surface temperature. The SWI variability at a given latitude is highest in the extratropics and polar regions with decreasing values equatorwards. This meridional distribution of SWI variability is explained by the variability in moisture source locations and its associated environmental conditions as well as transport processes. The westward located moisture sources of water vapour in the extratropics are highly variable in extent and latitude due to the frequent passage of cyclones and thus widen the range of encountered SWI values in

the marine boundary layer. Moisture loss during transport further contributes to the high SWI variability in the extratropics. In the subtropics and tropics, persistent anticyclones lead to well-confined narrow easterly moisture source regions, which is reflected in the low SWI variability in these regions. Thus, the expected range of SWI signals at a given latitude strongly depends on the large-scale circulation. Furthermore, the ACE SWI time series recorded at different heights above the ocean

surface provide estimates of vertical SWI gradients in the lowermost marine boundary layer. On average, the vertical gradients





with height found during ACE are $-0.1‰\,\mathrm{m}^{-1}$ for $\delta^{18}\mathrm{O}$, $-0.5‰\,\mathrm{m}^{-1}$ for $\delta^{2}\mathrm{H}$ and $0.3‰\,\mathrm{m}^{-1}$ for d-excess. Careful calibration and post-processing of the SWI data and a detailed uncertainty analysis provide a solid basis for the presented gradients. Using sea spray concentrations and sea state conditions, we show that the vertical SWI gradients are particularly large during high wind speed conditions with increased contribution of sea spray evaporation or during low wind speed conditions due to weak

vertical turbulent mixing and dominating effects from non-equilibrium fractionation. Although further SWI measurements at a higher vertical resolution are required to validate these findings, the simultaneous SWI measurements at several heights during ACE show the potential of SWIs as tracers for vertical mixing and sea spray evaporation in the lowermost marine boundary layer.

## 1   Introduction

The atmospheric water cycle is an essential component of the Earth's climate system. Its short-term variability is directly linked to our daily weather, including the occurrence of clouds and precipitation. The main source for atmospheric water in oceanic regions is ocean evaporation which is strongly influenced by the large-scale atmospheric flow (Aemisegger and Papritz, 2018) as well as small-scale turbulent and convective mixing (Risi et al., 2019). Ocean evaporation feeds moisture into the marine boundary layer (MBL), where the evaporated ocean water undergoes convective and turbulent mixing. The measurement of

surface evaporation fluxes over the ocean is difficult and moisture source attribution of MBL water vapour cannot be done by traditional atmospheric humidity measurements. A useful tool to investigate the influence of dynamical processes on the MBL water budget at various spatial and temporal scale are stable water isotopologues (SWIs, hereafter referred to as isotopes for simplicity). In this study, we investigate synoptic driving mechanisms of SWI signals in the MBL at different latitudes in the Atlantic and the Southern Ocean.

SWIs are usually quantified by the $\delta$-notation (Craig, 1961): $\delta[‰] = \left(\frac{R}{R_{\mathrm{VSMOW2}}} - 1\right) \cdot 1000$, where $R$ is the isotopic ratio of either $\mathrm{H}_2^{18}\mathrm{O}$ or $^2\mathrm{H}^1\mathrm{H}^{16}\mathrm{O}$ (with $R$ representing the ratio of the concentration of the heavy molecule to the concentration of $\mathrm{H}_2^{16}\mathrm{O}$). The $\delta$-notation expresses the relative deviation of the isotopic ratios $R$ from the internationally accepted primary water isotope standard, that is, the Vienna standard mean ocean water (VSMOW2; with $^2R_{\mathrm{VSMOW2}}$=$1.5576\cdot10^{-4}$ and $^{18}R_{\mathrm{VSMOW2}}$=$2.0052\cdot10^{-3}$). SWIs are tracers of moist atmospheric processes because they record phase changes in the atmo-

sphere. Whenever a phase change occurs, the relative abundance of SWIs is altered by isotopic fractionation. The difference in saturation vapour pressure between heavy and light isotopes causes isotopic fractionation, referred to as equilibrium fractionation, the strength of which is inversely related to temperature. A second type of fractionation, the non-equilibrium fractionation (Dansgaard, 1964; Craig and Gordon, 1965), occurs additionally to equilibrium fractionation if the two phases are not in equilibrium. This is the case, for example, during ocean evaporation. During non-equilibrium conditions, a net transfer of water

molecules occurs, whereby diffusion effects alter the relative abundance of SWIs due to the different diffusion velocity of the different water molecules. The secondary isotope variable deuterium excess ($d = \delta^2\mathrm{H} - 8\cdot\delta^{18}\mathrm{O}$; Dansgaard 1964) provides a measure of non-equilibrium fractionation. $d$ is close to zero in the absence of non-equilibrium effects at temperatures of around $20\,^\circ\mathrm{C}$. The mean global $d$ of water evaporated from the ocean is approximately $10‰$, which indicates that, on average,



non-equilibrium conditions are expected during evaporation (Craig, 1961).

Isotopic fractionation and the distribution of SWIs in the hydrological cycle have been studied since the early 1950s using measurements and modelling of SWIs (Epstein and Mayeda, 1953; Dansgaard, 1954; Craig, 1961, see also reviews of Gat, 1996, Galewsky et al., 2016). Commercially available cavity ring-down laser spectrometers have enabled an increasing amount

of field studies measuring SWIs during the past decade. These continuous, high-resolution measurements of SWIs in water vapour provide the necessary precision and accuracy to study short-term variability in moist processes at the timescale of typical weather systems (Aemisegger et al., 2012). Ship-based measurements of SWIs in water vapour have proven to be useful to identify governing processes that define the MBL moisture budget such as the influence of the advection of terrestrial air masses (Gat et al., 2003), the organisation of convective systems in the tropics (Kurita, 2013), mixing with the free troposphere in the

subtropics (Benetti et al., 2014), the movement of atmospheric fronts in extratropical and polar regions (Kurita et al., 2016), the sublimation of snow on sea ice in polar regions (Bonne et al., 2019) and the influence of relative humidity and sea surface temperature at diverse latitudes (Uemura et al., 2008; Bonne et al., 2019). Recent studies of SWI measurements in the atmospheric water cycle identified the importance of these processes mainly for specific regions. In extratropical and polar regions, the passage of atmospheric fronts leads to abrupt changes in air masses and strongly contrasting isotopic signatures ahead and

behind the front in precipitation (Gedzelman and Lawrence, 1990) and water vapour (Aemisegger et al., 2015; Kurita et al., 2016). Furthermore, moisture source locations and moisture transport paths are highly variable and depend strongly on the observation location in extratropical and polar regions (Steen-Larsen et al., 2013, 2015). In the subtropics, MBL air masses are prone to mixing between descending mid- to upper tropospheric air masses and water vapour from ocean evaporation, which leads to MBL isotope signals that are more depleted than the water vapour formed from ocean evaporation (Noone et al., 2011;

Benetti et al., 2014, 2015). In the tropics, convection exerts a strong control on moist atmospheric processes and leaves distinct isotopic signals in tropical precipitation depending on the degree of organisation of convective systems, convective downdrafts and cloud top height (Lawrence et al., 2004; Bony et al., 2008; Torri et al., 2017). Below-cloud interaction of isotopically depleted rain droplets with MBL water vapour also affects the SWI composition of the MBL. Deep convective rainfall in the tropics leads to a depletion of the SWIs in MBL water vapour compared to the isotopic signal from only ocean evaporation

(Lawrence et al., 2004; Kurita, 2013). Even though different processes have been identified at different latitudes, the relative importance of these processes for the isotopic composition in the MBL at different latitudes and in different large-scale flow configurations has not been assessed so far.

The MBL has been described by Brutsaert (1965) with a three-layer model. A viscous sublayer of a height of several millimetres that is in equilibrium with the sea surface is overlaid by a surface layer with tens of meters height, which is dominated

by turbulence, and, above, the well-mixed Ekman layer spans to the top of the MBL (Lewis and Schwartz, 2013). Ship-based measurements are normally situated in the surface layer and thus directly influenced by turbulent conditions. The effect of turbulence on isotopic fractionation is not well understood and normally assumed to be negligible in isotopic models of ocean evaporation (Horita et al., 2008; Feng et al., 2019). This classical view of a viscous sublayer that is strictly separated from the turbulence dominated layer above, has been questioned in a study where coherent structures were seen within turbulent flows

in water channel experiments (Tanny and Cohen, 2008). Within these coherent structures, laminar flow might be present and





diffusion processes could lead to fractionation effects of SWIs during the transport. Thus, the assumption of negligible isotopic fractionation in the turbulent layer might not be adequate under all conditions. These potential diffusive fractionation effects in the turbulent layer near the surface for different wind conditions have not been investigated so far, due to the lack of suitable measurements.

Atmospheric turbulence in the vicinity of the ocean surface is induced by momentum fluxes that depend on the sea state and wind speed. The sea state can be described by the wave age, the ratio of wind speed to group velocity [i.e. the speed at which the wave energy travels (Young, 1999)]. The wave age describes the ability of waves to absorb energy from the wind. When waves are young (wave age of a value $\sim< 1.0$), they travel slower than the wind and absorb energy from it; when waves are mature, they travel faster than the wind and no longer absorb energy from it. Young waves are normally characterised
by a steep profile and, hence, are more prone to breaking. The sea state does not only affect turbulence but also sea spray production. Sea spray is produced by breaking waves and bubble bursts which mainly occur during rough sea states at high wind speeds (Monahan et al., 1986) and young wave age. The production and subsequent evaporation of water from sea spray particles for a rough sea state introduces isotopically enriched water vapour into the lower MBL if the water from sea spray particles evaporates nearly completely. This process of water evaporating from sea spray particles will be refered to as sea
spray evaporation in the following. Gat et al. (2003) estimated that up to 50% of the measured humidity in the Mediterranean MBL can originate from sea spray evaporation. It is still an open question to what extent sea spray evaporation affects moisture in the MBL (Veron, 2015) in different large-scale wind forcing conditions. Due to the specific isotopic signature of sea spray evaporation, SWI measurements near the ocean surface might give further insight into the moisture contribution of this process.

In summary, ship-based measurements of SWIs in water vapour are influenced by processes acting at various spatio-temporal scales. To investigate these processes, recent studies focused mainly on specific regions and have not compared the relative importance of these processes at different latitudes. Furthermore, there is still a lack of measurements to study small-scale turbulent processes close to the ocean surface that could influence MBL moisture significantly, e.g. by sea spray evaporation. The objectives of this study are to investigate 1) the variability of SWIs in the oceanic MBL at different latitudes, 2) the
large-scale circulation drivers of SWI signals in different climate zones, and 3) the local small-scale drivers of SWI signals such as turbulent mixing and sea spray evaporation. This study combines the water vapour measurements from three cavity ring-down spectrometers at two different heights on a research vessel during the Antarctic Circumnavigation Expedition (ACE) in the Atlantic and the Southern Ocean in 2016/2017 to analyse the meridional and vertical SWI variations in this five-month dataset. The drivers of SWI variability in the MBL are identified with a special focus on the diagnosed Lagrangian moisture
sources (Sodemann et al., 2008). The Lagrangian perspective allows to study the conditions in the "catchment area" of MBL moisture and, thus, to compare local and remote drivers of SWI variability in the MBL. The study is structured in the following way: First, the measurement setup and calibration procedures are described (Section 2). Second, a detailed analysis of the difference between the datasets is given to assess possible measurement errors and calibration uncertainties (Section 3). Third, the variability of the SWI time series is analysed by presenting and discussing the dominant drivers for meridional and vertical
SWI variations (Section 4).



## 2 Methods and Data

In this section, measurements conducted during the Antarctic Circumnavigation Expedition (ACE, see Section 2.1) are described. The main dataset of this study are the ship-based measurements of SWIs in water vapour at two elevations on the research vessel with different measurement setups and independent calibration and post-processing procedures (Sections 2.2

and 2.3). Thereafter, the additional measurements, model datasets and methods of this study are described (Section 2.4).

### 2.1 Expedition

The Antarctic Circumnavigation Expedition (ACE) took place between 21 November 2016 and 11 April 2017 on the RV *Akademik Tryoshnikov* (Walton and Thomas, 2018). The expedition was divided into three main legs covering the circumnavigation of Antarctica in the Southern Ocean [legs 1-3] and two additional legs with Atlantic Ocean transects from Bremerhaven

(Germany) to Cape Town (South Africa) [leg 0] and back [leg 4] (see cruise track in Fig. 1). Amongst the wide range of observations during ACE, a comprehensive set of atmospheric in situ measurements of aerosol characteristics (Schmale et al., 2019) and SWIs were conducted. Two instrumentation setups for measuring SWIs in water vapour using Picarro laser spectrometers were installed on the RV *Akademik Tryoshnikov* (Fig. 2). One setup was installed approximately 8 m a.s.l. (hereafter refered to as SWI-8) and was measuring with two instruments on both sides of the vessel (port side (ps) and starboard side (sb)). The

second setup was situated at a height of approximately 13.5 m a.s.l. (hereafter refered to as SWI-13). Further atmospheric and oceanographic measurements were conducted during ACE on-board the RV *Akademik Tryoshnikov* providing the following datasets used in this study: atmospheric chemistry measurements next to the SWI-13, automated meteorological measurements, and remote sensing of wave activity (see Section 2.4). In the following, 1-hourly means of the measured datasets are shown with the 1-hourly standard deviations of the 1-second resolution time series, if not mentioned otherwise.

### 2.2 SWI-13

The SWI-13 measurements were conducted in the atmospheric measurement container on deck 2 of RV *Akademik Tryoshnikov*, approximately 13.5 m a.s.l. during legs 1 to 4 (Fig. 2). A custom-made plexiglass inlet was mounted on top of the container (see supplementary Fig. S1). Three layers of perforated plexiglass surrounded the inlet line to prohibit sea spray from reaching the line. The outermost layer was heated with a self-limiting heating band to avoid icing on the plexiglass surface of the inlet

and to hold the inlet temperature above ambient temperature to avoid condensation. A 1.5 m long heated PFA tube with 10 mm inner diameter connected the inlet with the laser spectrometer inside the container. A filter (0.2 $\mu$m PTFE vent filter) was used to prevent particles from entering the line. The heated PFA inlet line had a constant temperature of 50 °C and was flushed with a KNF pump at a pumping rate of 9 $\ell$ min$^{-1}$ leading to a total renewal of the air in the inlet line every second. A Picarro cavity ring-down spectrometer L2130-i was connected to the inlet line and operated continuously inside the container with a flow rate

through the cavity of 300 m$\ell$ min$^{-1}$.

Due to power issues, the SWI-13 measurement system had to be moved to the upper bridge at a height of 24.4 m a.s.l. during leg 0 (see Fig. 2, green triangle). Less sea spray was expected due to the increased height, and therefore a downward facing





teflon funnel, instead of the plexiglass inlet, was mounted to the inlet line.

During legs 1-4, the temperature inside the measurement container was regulated to $20 \pm 5\,°C$. In the tropics on legs 0 and 4, the temperature in the atmospheric measurement container and the room on the upper bridge exceeded $40\,°C$ which might have affected the measurements (see Section 3.1). Furthermore, some precipitate remained on the container roof after precipitation

events and short-term contributions of isotopically depleted moisture from precipitation evaporation to the measured air cannot be ruled out after such events. The inlet was inspected frequently and snow around the inlet was removed on a few occasions during leg 2.

Measurements with a known standard (calibration runs) were performed with an automated schedule using a standard delivery module (SDM) from Picarro. The L2130-i raw measurements were calibrated using the SDM calibration runs using a similar

procedure as described in Aemisegger et al. (2012): First, the data was corrected for the humidity dependent isotope bias [referred to as isotope-humidity dependency (e.g. Schmidt et al., 2010; Aemisegger et al., 2012; Steen-Larsen et al., 2013)] for all measurements with a water vapour mixing ratio below 12'000 ppmv. A few recent studies (Bailey et al., 2015; Bonne et al., 2019) showed that the isotope-humidity dependency of their instrument is additionally sensitive to the isotopic composition of the used standard. No such impact was found for our L2130-i optimised for higher flow rates (see supplementary Fig. S5).

Second, a two-point slope correction and normalisation to VSMOW2-SLAP2 using a 10-day running mean of the calibration runs was applied to correct for the drift of the instrument during the cruise. The calibration protocol and the isotope-humidity dependency correction are discussed in more detail in the supplementary material.

The water vapour mixing ratio $w$ measured by L2130-i was calibrated using a dew point generator (LI-COR LI 610). Before and after the cruise, calibration measurements were conducted in the lab with controlled mixing ratios between 5'000 and

32'000 ppmv.

## 2.3 SWI-8

The second set of SWI measurements in water vapour were conducted in the hydrological lab on the main deck. Two inlet lines were installed to measure on both sides of the research vessel at a height of approximately 8 m a.s.l. A Picarro cavity ring-down

spectrometer L2120 was connected to the portside inlet line (SWI-8-ps) during legs 0 to 4. For legs 2 and 3, a second Picarro laser spectrometer L2130-i was installed to measure on the starboard side (SWI-8-sb). The inlet lines were made of a ¼" copper tubing, which was isolated and constantly heated to $50\,°C$ in order to avoid the condensation of water vapour. The inlets were protected with a plastic bottle (see Fig. 2) and inspected several times per day.

Two types of calibration devices were used. A self-made device described in Steen-Larsen et al. (2014) was used on all legs

while a Picarro SDM was additionally used during legs 2-3. The SDM was used for the L2130-i calibration runs and the self-made device for both laser spectrometers. The same calibration routines were used as described for SWI-13 (see supplementary material) by applying a correction for the isotope-humidity dependency for all measurements with a water vapour mixing ratio below 15'000 ppmv and by correcting the instrument's drift with a two-point slope correction and normalisation to VSMOW2-SLAP2 using 10-day running means of the calibration runs for L2120 and 14-day running means for L2130-i, because the



calibration runs are available at a lower frequency for L2130-i. As the SDM was only used during two of the five legs, the dataset was calibrated using the self-made device, while the SDM outputs were used for comparison.

We tested the sensitivity of the final SWI-13 and SWI-8 time series to the calibration procedure. Different calibration versions were calculated by altering the calibration procedure. A detailed description and comparison of these different calibration versions is given in Section 3.

## 2.4 Additional measurement and model datasets

### 2.4.1 Atmospheric microphysical and chemical measurements: Sea spray concentration and exhaust mask

A sea spray proxy was calculated from the particle number size distribution obtained by an aerodynamic particle sizer (APS, TSI Model 3321), which was operated inside the atmospheric measurement container (Schmale et al., 2019). Here, we define the sea spray proxy as particles with a diameter larger than 700 nm (N700) for legs 1-3. In the Southern Ocean along the ACE cruise track, other sources of particles larger than 700 nm are negligible (Schmale et al., 2019). The sea spray proxy strongly underestimates the total number of particles originating from sea spray, since most of them are smaller. However, for our purposes, N700 is a good indicator to identify the influence of sea spray on the SWI measurements. No sea spray proxy was calculated for leg 4 where mineral dust and soot from forest fires also influenced the measurements and interfered with the identification of sea spray using just a particle diameter and no chemical information. No measurements are available for leg 0. The particle number size distribution measurements were influenced by the vessel's exhaust plume, depending on wind direction, wind speed and vertical atmospheric stability. The $CO_2$ mixing ratio, black carbon mass concentration and particle number concentrations show distinct signals during exhaust influence and were used to generate an exhaust mask, with which the sea spray proxy is cleansed. No influence by the exhaust plume on the SWI measurements was observed (see Section 3.2). Therefore the exhaust mask is not applied to the SWI time series.

### 2.4.2 Ocean surface state measurements

The sea state was continuously monitored using the wave and surface current monitoring system (WaMoS-II, Ziemer and Günther 1994; Dittmer 1995). WaMoS-II is composed of an analog-to-digital converter and a processing software to acquire and analyse video signals from the marine X-band radar on board of the RV *Akademik Tryoshnikov*. Standard image processing techniques based on Fourier transforms are used to extract the wave energy spectrum from which wave characteristics are derived to calculate the wave age (for details see supplementary material).



### 2.4.3 Meteorological measurements

An automated weather station (model: AWS420, Vaisala) was operated on the RV *Akademik Tryoshnikov* during ACE delivering measurements of air pressure at 20 m a.s.l., air temperature, dew point temperature and relative humidity at 23.7 m a.s.l., and relative and absolute wind speed and direction at 30 m a.s.l. The recorded measurements were processed automatically by

the Vaisala system. Dew point temperature, air temperature, and atmospheric pressure are used to calculate the specific humidity $q$. The relative humidity with respect to sea surface temperature ($h_{\mathrm{SST}}$) is defined as $h_{\mathrm{SST}} = \frac{q}{q_{sat,SST}}$, where $q_{sat,SST}$ is the saturation specific humidity at sea surface temperature. Calibrated sea surface temperature (SST) measurements from ACE using a thermosalinograph (Aqualine FerryBox by Chelsea Technologies Group Ltd.) are not yet available. Therefore, the SST from the European Centre for Medium Range Weather Forecasts (ECMWF) operational data (see Section 2.4.4) is interpolated

along the ship track and used to calculate $h_{\mathrm{SST}}$. In order to quantify the bias caused by airflow distortion due to the ship's superstructure, the observed relative wind speed was compared to the expected relative wind speed based on ECMWF analysis data (as described in Section 2.4.4) and a corrected true wind speed at 10 m above sea level was derived (for details on the wind speed correction, see supplementary material).

### 2.4.4 Model data and Lagrangian methods

The Lagrangian analysis tool LAGRANTO (Wernli and Davies, 1997; Sprenger and Wernli, 2015) was used to calculate 10-day air parcel backward trajectories using the three-dimensional wind fields from the six-hourly global operational analysis data of the ECMWF and short-term forecasts in between the analysis time steps, i.e. at 03, 09, 15, 21 UTC. The ECMWF fields were interpolated on a regular horizontal grid of $0.5°$ horizontal spacing on 137 vertical levels. Up to 56 trajectories were launched every hour from the surface to 500 hPa in steps of 10 hPa, and with increased vertical resolution in the lowest 20 hPa above sea

level, starting trajectories at 1, 2, 3, 4, 5, 10, 15 and 20 hPa above sea level.

The moisture sources of the MBL water vapour along the ACE ship track were calculated hourly using the Lagrangian moisture source diagnostic by Sodemann et al. (2008), adapted for identifying the sources of water vapour instead of precipitation (Pfahl and Wernli, 2008) based on the 10-day backward trajectories. The mean global atmospheric moisture residence time is 4-5 days with maximum residence time up to 8 days in polar regions and the eastern tropical Atlantic ocean (Läderach and Sodemann,

2016). Therefore, 10-day backward trajectories are expected to cover the moisture source areas along the ACE track. The mean source conditions (latitude, longitude, air temperature, and specific humidity) are calculated, weighted by the amount of moisture uptake. For each hour along the ACE track, the 75 % moisture source area is calculated. This area represents the source region of 75% of the total moisture at the measurement position neglecting the 25 % sources with the lowest moisture contributions. A moisture uptake-to-loss ratio (following Suess et al., 2019, adapted for water vapour in the MBL) is used as a

measure to compare the cumulative moisture uptake to the cumulative rain out of air parcels in the MBL at the measurements site. Changes in the specific humidity $q$ for each timestep during the 5 days prior to arrival along the backward trajectories starting within the MBL are used to calculate the ratio. An increase in $q$ within a timestep is interpreted as an uptake of moisture by the air parcel, decreasing $q$ as a loss of moisture. A high uptake-to-loss ratio represents low moisture loss relative



to moisture uptake during the 5 days before arrival and, thus, minor influence by rain out on the measured SWI composition is expected.

Cyclone frequencies were calculated by applying a 2D cyclone detection algorithm, which identifies the outermost closed sea level pressure contour which encloses a pressure minimum (Wernli and Schwierz, 2006; Sprenger et al., 2017) using the

ECMWF operational analysis data. Accordingly, anticyclone frequencies were calculated using the outermost closed sea level pressure contour which encloses a pressure maximum.

## 3    Uncertainties from the SWI post-processing procedure

To identify robust deviations between the measured SWI time series from different locations on the ship, uncertainties due to the measurement and post-processing procedure are assessed. This allows for a quality check of the SWI time series to identify

robust small-scale horizontal and vertical differences in SWIs in the lowermost MBL. The comparison of SWI-8-ps and SWI-8-sb provides a measure of horizontal variations in SWIs around the research vessel, whereas the comparison of SWI-8-ps with SWI-13 gives an estimate of vertical variations in SWIs (compare Fig. 2). The difference between SWI-8-sb and SWI-8-ps has a mean value of $0.8\,[-1.6\,...\,3.2]\,‰$ for $\delta^2$H (numbers in brackets denote the 65% percentile range), $-0.04\,[-0.41\,...\,0.38]\,‰$ for $\delta^{18}$O and $1.2\,[-0.2\,...\,2.4]\,‰$ for $d$ (see also Appendix Fig. A1) and are smaller than the vertical differences (Fig. 5). Large

horizontal differences between SWI-8-sb and SWI-8-ps are observed only during short time periods, most likely due to sea spray influence on one of the two sides. The horizontal differences can be interpretated as the expected noise due to small differences in the measurement setup (e.g. length of inlet line, angle of inlet towards the ocean surface, ship's structure at inlet position). In the following, we will use SWI-8-ps to represent the measurements at 8 m a.s.l. because these measurements are available during the entire expedition. The vertical differences between SWI-13 and SWI-8-ps ($\Delta_{13-8}$) over all legs are up to

an order of magnitude larger than the horizontal differences for the $\delta$-values, with $-2.6\,[-4.8\,...\,-0.2]\,‰$ for $\delta^2$H (numbers in brackets denote the 65% percentile range), $-0.55\,[-0.90\,...\,-0.14]\,‰$ for $\delta^{18}$O and $1.8\,[0.5\,...\,3.2]\,‰$ for $d$ (for details see also Appendix Fig. A1). The robustness of these vertical differences is assessed in the following uncertainty analysis, which focuses on the effects of instrument properties, pollution by the ship's exhaust and the calibration procedure.

### 3.1    Instrument properties

High quality laser spectrometric measurements rely on a precise regulation of temperature and pressure within the instrument's cavity. The target cavity pressure (CP) is regulated to $50\pm0.02$ and $35\pm0.03$ Torr, for SWI-13 and SWI-8-ps respectively, and the cavity temperature (CT) to $80\pm0.002\,°$C. To exclude differences in the SWI signal due to an unstable cavity environment, data points were excluded if the cavity pressure and temperature deviated by more than 0.2 Torr and $0.02\,°$C, respectively, from

the target cavity pressure and temperature (0.2 % of all data points for SWI-13 and 0.6% for SWI-8-ps). The remaining data points are analysed for a potential dependency of $\Delta_{13-8}$ on the cavity properties. The vertical differences in $\delta$-values between SWI-13 and SWI-8-ps, $\Delta_{13-8}\delta^2$H and $\Delta_{13-8}\delta^{18}$O, do not show any correlation with deviations from the target cavity prop-





erties of the respective instruments, with a Pearson correlation coefficient smaller than 0.1 for $\Delta_{13-8}\delta^2$H resp. $\Delta_{13-8}\delta^{18}$O correlated with CP or CT of each SWI-13 and SWI-8-ps. Also during the high temperatures at the measurement site in the tropics, the cavity environment does not show any irregularities. Thus, variations in the cavity environment do not contribute to the differences between SWI-13 and SWI-8-ps. A detailed analysis of the observed variations in cavity properties is given
in the supplementary material.

### 3.2 Influence of exhaust air

Chemical measurements at the inlet site of SWI-13 showed episodic pollution by the vessel's exhaust air (see section 2.4.1). Exhaust air might affect the SWI measurements in water vapour by altering the ambient air's gas mixture, and by the presence
of e.g. hydrocarbons impacting the spectroscopic baseline (Aemisegger et al., 2012; Johnson and Rella, 2017). A possible exhaust impact on the SWI measurements was analysed by studying $\Delta_{13-8}$ during exhaust and no exhaust periods. The medians of the $\Delta_{13-8}$-distributions for $\delta^2$H, $\delta^{18}$O, and $d$ are shifted towards zero by 1.6‰, 0.3‰ and 0.7‰, respectively, for periods with exhaust influence relative to periods without (see Appendix Fig. A2a-c). The periods with exhaust influence are dominted by westerly winds (Appendix Fig. A2d) and, thus, the measurements during exhaust influence are mainly associated with
zonal advection. The dominance of this large-scale advection situation for the exhaust periods could be the main reason for the observed difference in the $\Delta_{13-8}$ distributions for periods with and without exhaust influence. Furthermore, large vertical differences occured more often during periods without exhaust influence and are thus unlikely to be caused by pollution from the ship's exhaust. Therefore, the exhaust influence on the SWI measurements is considered to be negligible and the exhaust masked is not applied to the SWI-13 time series.

### 3.3 Uncertainties in the calibration procedure

The influence of the various steps in the calibration protocol of SWI-13 and SWI-8-ps is assessed with sensitivity tests by varying one of the following steps and measuring the impact on the calibrated time series:

1. To correct the data for the isotope-humidity dependency, isotope-humidity dependency correction curves are derived
using least-square fits to the standard measurements at different water vapour mixing ratio. To estimate the uncertainty of these fitted correction curves, different isotope-humidity dependency correction curves are applied: The correction curves from ACE ($\mathcal{H}_1$ and $\mathcal{H}_3$), a minimum and maximum correction curve ($\mathcal{H}_{1,\min}$, $\mathcal{H}_{3,\min}$ and $\mathcal{H}_{1,\max}$, $\mathcal{H}_{3,\max}$) for the ACE data representing the best fit to the calibration runs $\pm 1$ standard deviation in $\delta$-values of the calibration runs, the correction curve from Sodemann et al. (2017) ($\mathcal{H}_2$) or no humidity correction ($\mathcal{H}_c$).

2. To correct for drifts between calibration runs, either a 10-day running mean is calculated from the runs or, for each leg, the average over all runs is used.





The calibration versions are summarised in Table A1 in the Appendix. In the following, the versions are compared to the final version which are calibrated using $\mathcal{H}_1$ and $\mathcal{H}_3$ for SWI-13 and SWI-8-ps, respectively, and a 10-day running mean for the drift correction between calibration runs. The isotope-humidity dependency correction (step 1) has the strongest impact on the calibration procedure. The uncertainty of the isotope-humidity dependency correction function, estimated by the minimum ($\mathcal{H}_{1,min}$, $\mathcal{H}_{3,min}$) and maximum ($\mathcal{H}_{1,max}$, $\mathcal{H}_{3,max}$) correction functions, leads to an uncertainty in the calibrated time series smaller than the mean 1-hourly standard deviation of 0.3 ‰, 2.3 ‰ and 2.8 ‰ for $\delta^{18}$O, $\delta^2$H and $d$, respectively, except for $\delta^2$H and $\delta^{18}$O at $w < 3'000$ ppmv. Varying the handling of the calibration runs (step 2) introduces small differences on the order of 0.2 ‰ and 0.1 ‰ for $\delta^2$H and $\delta^{18}$O, respectively. Comparing the variations due to different calibration procedures with the vertical variations in SWIs, we find that $\Delta_{13-8}$ is larger than the difference between the calibration versions, except for $\delta^2$H using the calibration of SWI-8-ps with $\mathcal{H}_c$ (no isotope-humidity dependency correction). However, for the other variables, the differences from the final version for the version without isotope-humidity dependency correction of SWI-8-ps correspond to less than 50% of $\Delta_{13-8}$. Adding the effect of the uncertainty of the isotope-humidity dependency correction curves (calculated from the minimum and maximum correction curves) of both SWI-13 and SWI-8-ps, changes in the calibrated time series amount to 53%, 75% and 39% of $\Delta_{13-8}$ for $\delta^{18}$O, $\delta^2$H and $d$, respectively. Even though, the isotope-humidity dependency correction introduces some uncertainty into the calibrated data, the latter remains distinctly smaller than $\Delta_{13-8}$ and, based on our current knowledge of factors influencing the calibration procedure, cannot fully explain the vertical differences in the SWI measurements. For more details on the calibration versions, see supplementary material.

We conclude, that measurement-related factors which could influence the SWI time series, such as instrument settings, exhaust influence and the calibration procedure, cannot explain the observed vertical differences between the two time series, $\Delta_{13-8}$. Thus, the vertical differences are considered to be robust and the natural processes driving them are further discussed in Section 4.2.

## 4    Results and Discussion

The five-month time series of SWIs in water vapour provide the unique opportunity to assess moisture source and transport processes in the MBL on various time scales and under diverse climatic conditions. In this section, the meridional and vertical variations of the SWI composition of the MBL are analysed. First, the time series are interpreted in their climatic context and meteorological processes responsible for the SWI variations along the meridional transect from 60 °N to 80°S are analysed. Second, it is illustrated how simultaneous SWI measurements at different heights can be used to study vertical isotope gradients and to estimate sea spray influence in the lowermost MBL.

The time series of SWI-13 and SWI-8-sb/ps correlate well with a Pearson correlation coefficient larger than 0.95 for all legs, instruments and variables, except for $d$ which shows a Pearson correlation coefficient between SWI-13 and SWI-8-ps of 0.9 for leg 0. $\delta^{18}$O and $\delta^2$H are lower and $d$ is higher in SWI-13 than in SWI-8-sb/ps for most of the time except for the tropics and parts of leg 3. Due to the high correlation between the times series, only SWI-13 is discussed in the following section. SWI-13





is chosen because calibrated measurements of $w$ are available for this dataset (see Section 2.2) and less sea spray influence is expected at the higher inlet location.

## 4.1 Meridional SWI variations

The diverse climate zones from the tropics to polar regions, traversed during the expedition, provide the possibility to probe a variety of different environmental conditions in various large-scale atmospheric forcing situations. The SWI measurements in Fig. 3 and 4 show a high event-to-event variability overlaid by a meridional gradient. The SWI values spread from $-8.6‰$, $-65.6‰$ and $20.3‰$ during leg 0 and 4 to $-37.1‰$, $-291.0‰$ and $-9.0‰$ during leg 2 and 3 for $\delta^{18}$O, $\delta^2$H and $d$, respectively. Here, we investigate the drivers of these meridional SWI variations and aim to disentangle the effect of short-term

synoptic events on SWI variability from the effect of varying climatic conditions.

### 4.1.1 Imprint of varying climatic conditions on SWI signals

To investigate the meridional SWI variations, the data was grouped into bins of $10°$ latitudinal width. Figure 6 shows boxplots of these bins for measured SWI and meteorological variables. Furthermore, boxplots of the weighted mean moisture source conditions (Figs. 6d-f) are shown. Even though the number of points per bin differs strongly between the extratropics in the

Northern and Southern hemisphere (not shown), the corresponding latitudinal bins of the two hemispheres cover a similar range of values for SWI and meteorological values.

The binned median $\delta^2$H and $\delta^{18}$O values show distinct meridional distributions with on average isotopically enriched air masses in the tropics and depleted air masses in polar regions (Figs. 6a,b). Note that the bin representing measurements closest to Antarctica shows an increase in median $\delta$-values compared to the adjacent bin to the North. The measurements between

$80°$S and $70°$S contain only observations from four continuous days, and therefore are strongly influenced by one weather situation.

A meridional gradient is also visible for the bins' interquartile ranges (refered to as IQR, and labeled by the subscript $_{\text{IQR}}$ in the following). The enriched environments in the tropics and subtropics show small IQRs in SWI variables, whereas the depleted extratropical to polar regions show large IQRs. In both hemispheres, $\delta^2$H$_{\text{IQR}}$ and $\delta^{18}$O$_{\text{IQR}}$ increase from $20° - 70°$ latitude.

The variability of SWIs is especially high in the $40 - 50°$ latitude band in both hemispheres. In this band, the [5,95]-percentile range extends over a similar range as the meridional gradient in median $\delta$-values between the extratropics and the tropics. The very large [5,95]-percentile range of SWI measurements in the band $60 - 70°$S is due to very low $\delta$-values measured at the Mertz glacier on 29 January 2017 (see Fig. 4), which leads to a strongly skewed distribution of measurements in this latitudinal band.

The binned environmental conditions at the measurement site give insight into potential reasons for the meridional SWI variations described above. Higher temperature ($T$) and specific humidity ($q$) at the ship's position and averaged over the moisture sources are observed in the tropics compared to higher latitudes (Figs. 6d,e). Both, the median $T$ and $q$, show similar meridional distributions as the median $\delta^2$H and $\delta^{18}$O. The distribution of $T$ is asymmetric with higher temperatures in the $30 - 40°$S band





compared to the same latitude in the Northern Hemisphere. This asymmetry in the $T$ distribution is reflected in the median $\delta^2$H distribution and expresses the seasonal contrast between the winter and the summer hemispheres. $T$ and $q$ are lower at the moisture source compared to the measurement site which reflects properties of air masses that experience moistening due to ocean evaporation and warming due to heat exchange with the ocean. They are initially cold and dry, and are advected

over a relatively warmer ocean, thereby triggering ocean evaporation by a strong humidity gradient between the ocean and the atmosphere (Aemisegger and Papritz, 2018). For the southernmost bin, the warmer $T$ at the moisture source compared to the adjacent bin to the North hints towards more equatorwards sources and transport within the warm sector of an extratropical cyclone, which can explain the relatively enriched SWI composition in this bin closest to Antarctica.

Similar meridional variations, as seen here for $\delta^{18}$O and $\delta^2$H in water vapour, have been observed for SWIs in precipitation

(Araguás-Araguás et al., 2000; Feng et al., 2009). These meridional SWI variations are interpreted traditionally as the isotopic depletion of air masses due to rain out and were described by Dansgaard (1964) as the "temperature effect". Previous measurements of SWI in water vapour in the Atlantic Ocean show a similar meridinal gradient with highest values in the tropics (around -10 ‰ for $\delta^{18}$O) and lowest in polar regions (around -35 ‰ for $\delta^{18}$O, Bonne et al., 2019). Close to the equator, Liu et al. (2014) observed slightly different patterns of meridional variations in their measurements of SWIs in water vapour from the Indian

Ocean. They showed a depletion in SWIs in water vapour in the tropics compared to the subtropics. This is probably due to the proximity of the southeast Asian land masses and the influence of deep convective precipitation systems in the tropics, conditions that were not encountered during the SWI measurements in the Atlantic ocean. There are a few periods of depleted $\delta^{18}$O in the tropics in the measurements by Bonne et al. (2019) (see their Fig. 1), similar to the precipitation event in the tropics during ACE, which led to a strong short-term isotopic depletion of the MBL by 12 ‰ in $\delta^{18}$O due to convective downdrafts and

below-cloud interaction of hydrometeors with MBL vapour (see leg 0 at 2°N in Fig. 3). Due to the rare occurrence of tropical rainfall along the ACE track, the direct SWI imprint of isotopically depleted rainfall in the tropics might be underestimated compared to climatological conditions.

The meridional distribution of $d$ (Fig. 6e) with a peak in the tropics at median values of ~15 ‰ and minima around 0 ‰ close to Antarctica is in line with the $d$ predicted from local $h_\mathrm{SST}$ and SST conditions using reanalysis data in Aemisegger and Sjolte

(2018) based on the closure assumption of Merlivat and Jouzel (1979). Meridional variations in $h_\mathrm{SST}$ at the measurement site are weak with higher $h_\mathrm{SST,IQR}$ at higher latitudes (Fig. 6f). The local $h_\mathrm{SST}$ measurements are anti-correlated with $d$, as shown in Fig. 7 as expected from detailed analysis of $h_\mathrm{SST}$ versus $d$ in water vapour from the Mediterranean (Pfahl and Wernli, 2008), the Southern Ocean (Uemura et al., 2008), and the Atlantic Ocean (Bonne et al., 2019). The meridional gradient observed in $d$ is partly due to the dependency of $d$ on SST during equilibrium fractionation and also reflects the meridional SST gradient as

discussed in Aemisegger and Sjolte (2018). The linear relations between $d$ and its environmental controls based on ACE data for evaporative conditions ($h_\mathrm{SST}$<100%) are $-0.4$ ‰ %$^{-1}$ and $0.4$ ‰ K$^{-1}$ for $h_\mathrm{SST}$ and SST, respectively. These values are consistent with previous ship-based studies (e.g. Uemura et al., 2008; Bonne et al., 2019). The SST-$d$ relationship from ACE is illustrated in Fig. 7. Several transient periods of high $d$ concurrent with low $h_\mathrm{SST}$ were observed along the Aghulas warm ocean current in the Southern Ocean (21 - 25 December 2016; "x" in Fig. 7) and along the sea ice edge albeit with lower peak $d$ values

for the same $h_\mathrm{SST}$ over regions with colder SSTs (13 - 16 February 2017; "+" in Fig. 7). In contrast to the observed positive lin-





ear correlation of $d$ and SST during ACE, results by Pfahl and Wernli (2008) and Steen-Larsen et al. (2015) showed only weak SST-$d$-correlations. This discrepancy might be due to the different spatio-temporal focus of these studies, which both used measurements from a fixed station, at the synoptic time scale, in an environment with weak SST gradients. The SST influence on $d$ was also questioned by Pfahl and Sodemann (2014). Their spatial $d$ distribution predicted using a combination of datasets

from the South Indian Ocean and the Mediterranean shows several marked differences to our results. In particular, $d$ measured during ACE is 5-10‰ smaller along the sea ice edge over low SST and 5-10‰ larger in the tropics over high SSTs than the DJF mean predicted by Pfahl and Sodemann (2014). The only exception in the decreasing meridional trend in $d$ are the elevated values in the northernmost bin, which groups measurements from the British Channel region. In this region, confined by land masses, we expect some influence of higher $d$ vapour from continental air masses, which have been moistened by evapotran-

spiration (Aemisegger et al., 2014). A few events with supersaturated conditions ($h_{\mathrm{SST}}$>100%) are associated with a moisture flux from the atmosphere into the ocean during warm air advection. These will be discussed in more detail in a follow-up study.

In summary, the meridional distribution of the $\delta^{18}$O, $\delta^2$H and $d$ signals can be linked to the varying climatic conditions, such as $T$, $q$, and $h_{\mathrm{SST}}$, along the ACE track, which are reflected in the median isotopic signature in the MBL water vapour. In the

15 next section, the large-scale dynamical drivers of the SWI signals from ACE are described.

### 4.1.2  Imprint of large-scale atmospheric weather systems on SWI signals

The meridional distribution of $\delta^{18}$O$_{\mathrm{IQR}}$, $\delta^2$H$_{\mathrm{IQR}}$, and $d_{\mathrm{IQR}}$ (Fig. 6b,d,f) is strongly linked to the type of weather systems involved in shaping the isotope signals on synoptic timescales. The drivers of the SWI variability at different latitudes are discussed in this section based on the moisture source properties and the frequency of occurrence of weather systems typical

for the traversed regions.

The meridional distribution of the weighted mean moisture source latitude (not shown) shows higher IQRs at higher latitudes. The large spread of moisture source locations in extratropical and polar regions is illustrated by the hourly 75% moisture source regions for the MBL water vapour along the cruise track (Fig. 8). The coloured contours in Fig. 8 represent the 75% moisture source region for locations in the same colour along the ACE track - the yellow contours in Fig. 8b, for example, correspond to

25 locations around 80 °E on the ACE track. For legs 1-3 (Fig. 8b-d), the moisture source regions cover nearly the whole Southern Ocean. The high temporal variability of the extratropical and polar moisture source areas is due to the high frequency of high and low pressure systems at these latitudes. The frequent passage of extratropical cyclones (Fig. 9) and their associated cold and warm sectors leads to an alternating SWI pattern by cold and warm advection (Dütsch et al., 2016; Aemisegger, 2018). A further common feature of the extratropics are the westerly moisture source regions relative to the ship's position, which are

30 due to the mean westerly winds and the eastward movement of extratropical cyclones within the storm track.

In contrast to the widespread moisture source areas in the extratropics, the source areas in the subtropics and tropics are narrowly confined. They extend in the direction of the trade winds (Fig. 8 a,e) and are located to the east and on the poleward side of the ship's position. The small $\delta^2$H$_{\mathrm{IQR}}$ and $\delta^{18}$O$_{\mathrm{IQR}}$ reflect the steady environmental conditions associated with these well-defined, narrow moisture source bands of the slowly subsiding subtropical air masses. In the tropics and subtropics, the



SWI variability in the MBL is dominated by vertical transport such as shallow and deep convection, turbulent mixing, and the influence of large-scale descending air masses in the subtropics (Lee et al., 2011; Brown et al., 2013; Benetti et al., 2015). During ACE, the flow conditions in the subtropics are dominated by low-level anticyclones, which lead to large-scale subsidence of air masses (Fig. 9b). These descending air masses are transported equatorward and experience extensive moistening due

to ocean evaporation and shallow convection in the MBL (Fig. 8). Due to the relatively stationary anticyclones and persistent trade winds in the subtropics during ACE, the moisture is transported along a north-east to south-western pathway with small temporal variability from the subtropics into the tropics. This persistent large-scale flow situation leads to similar moisture source locations for a given latitude throughout the tropics and subtropics and similar isotopic compositions of the evaporative flux at the moisture source. Therefore, small IQRs of SWIs have been observed in the tropics and subtropics. One exception is

the influence of North African air masses on $d$ which will be discussed later in this section.

In addition to the important role played by moisture source conditions, the measured SWI variations in water vapour can be further influenced by moisture removal and precipitation-vapour interactions during transport, both in the tropics (Lawrence and Gedzelman, 1996; Lawrence et al., 2004; Bony et al., 2008) and extratropics (Graf et al., 2019). These interactions modify the SWI composition of air masses during transport such that the measured isotopic composition deviates from the isotopic

composition of water vapour from ocean evaporation at the moisture source. The uptake-to-loss ratio, a measure of moisture uptake relative to moisture loss during transport (see Section 2.4.4), was 2-4 times larger in the subtropics and tropics than in the extratropics and polar regions (see Appendix Fig. A3). This implies that polar and extratropical air masses are more strongly affected by precipitation during transport than suptropical and tropical air masses, which reflects the highly dynamical nature of the atmospheric water cycle in the extratropics. Consequently, the high SWI variability of the MBL water vapour in

the extratropics is not only due to the strongly varying transport pathways and moisture sources, but also due to a larger degree of precipitation along these pathways. Note, that the fact that we did not encounter deep convective systems during this cruise leads to a high uptake-to-loss ratio in the tropics compared to what we expect in typical tropical deep convective regions.

The meridional variations of $d_{\mathrm{IQR}}$ (Fig. 6c) are more complex than the ones of $\delta^2\mathrm{H}_{\mathrm{IQR}}$ and $\delta^{18}\mathrm{O}_{\mathrm{IQR}}$. The smallest $d_{\mathrm{IQR}}$ are also found in the tropics, however a $d_{\mathrm{IQR}}$ maximum in the $10 - 20°$N band along the North African coast coincides with large

$h_{\mathrm{SST,IQR}}$ (see Fig. 6c and d for the corresponding latitudes in Fig. 3). These large $h_{\mathrm{SST,IQR}}$ and $d_{\mathrm{IQR}}$ reflect the strongly varying importance of African moisture source contributions and the contrasts in source locations between leg 0 and 4 (compare Figs. 8a and d). Except for the special case of the $10 - 20°$N band along the African coast, $d_{\mathrm{IQR}}$ is on average higher in the extratropics compared to the tropics with two local minima in regions of cold ocean surface currents at $50° - 60°$S in the region of the polar front and at $30° - 40°$N along the cold Canary surface ocean current.

The meridional distribution of SWI signals and their synoptic-timescale variability reveal the different driving processes at different latitudes. The meridional gradient of the median SWI composition in the MBL reflects the climatic conditions at the measurement and moisture source site, specifically $T$, $q$, and $h_{\mathrm{SST}}$. Dynamical drivers such as extratropical cyclones and persistent anticyclones control the variability of MBL SWI composition at a given latitude. The measured SWI signals, thus,





show an imprint of the environmental conditions in the MBL. The variability of these environmental controls and the measured SWI signals is driven by the dynamics of the large-scale circulation.

## 4.2 Vertical SWI variations

An estimate of the vertical SWI gradients in the near-surface layer is given by the difference between SWI-13 and SWI-8-ps signals ($\Delta_{13-8}$). Here, only the isotope variables $\delta^{18}$O, $\delta^{2}$H and $d$ are discussed, since calibrated specific humidity measurements are only available from SWI-13. More depleted water vapour was systematically measured at the 13.5 m site compared to the 8 m site on the research vessel. The influence of measurement uncertainties has been analysed in Section 3 and it has been shown that, to the best of our knowledge, they cannot explain the observed vertical differences. Therefore, physical reasons to explain the vertical SWI variations are discussed in this section. Only legs 1-3 are analysed because ocean surface state measures and sea salt concentrations are available solely for the Southern Ocean part of the cruise and because we expect different processes to affect the near-surface isotopic composition in the tropics compared to the extratropics. The data is shown in 5 min resolution in this section, because turbulence varies on sub-hourly timescales.

The vertical gradients sampled during ACE between the two measurement points at 13.5 m and 8 m a.s.l. amount to $-0.5\,[-0.9\ldots 0.0]\,‰\,\mathrm{m}^{-1}$ for $\delta^{2}$H (numbers in brackets denote the 65% percentile range), $-0.10\,[-0.16\ldots -0.02]\,‰\,\mathrm{m}^{-1}$ for $\delta^{18}$O, and $0.3\,[0.1\ldots 0.6]\,‰\,\mathrm{m}^{-1}$ for $d$ (Fig. 10) with overall more depleted vapour and higher $d$ at the higher elevation than closer to the sea surface. These gradients are approximately twice as large for $\delta^{18}$O and about the same order of magnitude for $\delta^{2}$H as the ones obtained from aircraft-based measurements in the MBL in the Mediterranean (Sodemann et al., 2017). Since the measurements during ACE were performed much closer to the surface, different vertical gradients can be expected. The vertical gradient in $d$ is opposite to the one observed by Sodemann et al. (2017). As influences by cloud processes, which might have played an important role in shaping the negative vertical $d$ gradient in Sodemann et al. (2017), can be neglected for near-surface measurements in the absence of fog, a positive vertical gradient for $d$ in the ACE measurements most likely reflects effects of non-equilibrium fractionation during evaporative conditions. Vertical SWI gradients measured in the Mediterranean between 20.35 and 27.9 m a.s.l. on a research vessel are of an order of magnitude smaller for $\delta^{18}$O and about the same order of magnitude for $d$ with large positive gradients in $d$ ($0.6\,‰\,\mathrm{m}^{-1}$) in the Eastern Mediterranean during conditions with high stability of the air column (Gat et al., 2003). Furthermore, the best representation of the vertical SWI gradients in idealized box models was achieved if the box models included sea spray evaporation (Gat et al., 2003). We will study the influence of vertical atmospheric stablity and sea spray evaporation on vertical SWI gradients further by considering in situ measurements of sea spray and wave age.

### 4.2.1 Sea spray and wave age

The dependency of $\Delta_{13-8}$ on local environmental conditions is examined to identify the processes that shape the vertical SWI gradients. The observed vertical gradients show a wind dependency with larger negative $\Delta_{13-8}\delta^{2}$H and $\Delta_{13-8}\delta^{18}$O for high wind speeds and, to a smaller extent, for very low wind speeds compared to intermediate wind speeds (Fig. 10). $d$ shows a weaker wind dependency than $\delta^{18}$O and $\delta^{2}$H with largest positive $\Delta_{13-8}d$ for low wind speeds. According to this dependency,





three wind regimes are defined: [I] low wind speed $< 6\,\mathrm{m\,s^{-1}}$, [II] intermediate wind speed between $6\,\mathrm{m\,s^{-1}}$ and $16\,\mathrm{m\,s^{-1}}$, [III] high wind speed $> 16\,\mathrm{m\,s^{-1}}$. Regime III shows the most extreme vertical SWI gradients, except for $\Delta_{13-8}d$, for which regime I shows the largest vertical gradients (see Appendix table A2). The large $\Delta_{13-8}$ in $\delta$-values in regime III coincide with high sea spray concentrations at the upper inlet (Fig. 10a-c). Sea spray influences SWIs by sea spray evaporation. Under the

assumption that water from sea spray droplets evaporate nearly completely, part of the moisture input into the MBL occurs through a non-fractionating process with signals close to the ocean surface isotope composition. A stronger influence of sea spray evaporation at the lower compared to the upper inlet could lead to a more enriched isotopic composition of water vapour at the lower inlet and thus large negative $\Delta_{13-8}\delta^2\mathrm{H}$ and $\Delta_{13-8}\delta^{18}\mathrm{O}$ at high wind speed. Because ocean water has a $d$ close to zero, the evaporation signal from sea spray introduces a low $d$ at the lower level and, thus, an increased positive $\Delta_{13-8}d$ is

expected. There is a weak tendency to larger positive $\Delta_{13-8}d$ in regime III, but the median $d$-values are close to the median values in regime II. It cannot be ruled out, that sea spray droplets were deposited on the inlet filter, where they evaporated. This measurement artefact is expected more frequently at the lower inlet and could enhance the isotopic signal induced by sea spray evaporation. As a measure of sea surface roughness and the production of sea spray from wave breaking, wave age is shown in Figs 10d-f. Wave age decreases with increasing wind speed. There are very few wave age measurements available at very high

wind speeds during ACE. The available wave age data suggest that the large negative $\Delta_{13-8}\delta^{18}\mathrm{O}$ and $\Delta_{13-8}\delta^2\mathrm{H}$ in regime III with high sea spray concentrations occurred during breaking wave conditions at low wave age ($\sim < 1.0$). Thus, the large vertical SWI differences in regime III can be explained by sea spray influence, which is stronger at the lower inlet location. For regime I at low wind speed, wave age is high and a much weaker influence of sea spray evaporation is expected. During these calm condition, there is again a tendency for larger negative $\Delta_{13-8}$ in $\delta$-values and larger positive $\Delta_{13-8}d$ compared to regime

II. These increased vertical SWI differences in regime I might be caused by weak vertical turbulent mixing and comparatively strong diffusive effects. This hypothesis is elaborated in more detail in section 4.2.2.

### 4.2.2   Effects of marine boundary layer turbulence

Here, an attempt is made to explain the observed near-surface vertical SWI gradients in the extratropics from a process-based perspective. Our analysis focuses on the atmospheric layer close to the ocean surface with a width of several tens of meters.

Note that, in contrast, the vertical mixing model introduced by Benetti et al. (2018) focused on vertical moisture mixing across the MBL top. We propose a qualitative interpretation framework based on the near-surface wind speed and the roughness of the sea surface (Fig. 11). Three processes are taken into account in this framework. First, in a hypothetically wind- and turbulence-free atmosphere without steady-state conditions, the vertical moisture gradient close to the ocean surface induces a diffusional upward moisture flux, which leads to an isotopic depletion and an increase in $d$ with distance from the ocean surface due to

non-equilibrium fractionation. Second, vertical turbulent mixing, which increases with wind speed, leads to a well-mixed layer close to the ocean surface and, thus, weakens the vertical SWI gradients. Third, the sea state determines the production of sea spray and the influence of sea spray evaporation on SWI composition. The proposed framework again considers the three wind regimes introduced in the previous section, in which these three processes are expected to differ in strength and therefore exert a varying influence on the vertical SWI gradient in the lowermost MBL.





First, for low wind speed conditions with high wave age in regime I (Fig. 11a), weak vertical turbulent moisture transport is expected. This leads to a weakly mixed MBL with a strong vertical moisture gradient. Due to this moisture gradient, non-equilibrium fractionation leads to stronger isotopic depletion and higher $d$ with higher elevation. This stratification is only moderately reduced by the weak turbulence. An enrichment effect due to sea spray is not expected in this case as only little sea spray is produced at low wind speeds. The combination of these processes could explain the measured vertical SWI structure for low wind speeds. Second, for intermediate wind speeds in regime II (Fig. 11b), turbulent mixing is stronger, which weakens a potential vertical SWI gradient induced by diffusion and leads to a well-mixed surface layer. In this regime, the influence by sea spray evaporation is still considered small. Therefore, small vertical SWI differences are measured. Other studies also showed an increasingly well-mixed lower MBL for lower wave age using vertical wind profiles in conditions representative of regime II (e.g. Smedman et al., 2009). Third, for high wind speeds in regime III (Fig. 11b), strong turbulent mixing is assumed in the lower MBL and sea spray production is enhanced which increases the vertical gradient in SWIs as shown in the previous section.

Even though this qualitative framework can explain the observed vertical SWI differences, it remains difficult to quantify the relative importance of enhanced turbulence and sea spray evaporation from two vertical point measurements. In particular, regime I with weak vertical turbulence at low wind speeds needs to be assessed in more detail including vertical profiles of specific humidity measurements. Therefore, more high-resolution vertical profiles of the SWI composition and environmental parameters such as temperature, specific and relative humidity, sea salt concentrations, and 3D wind speed in the lowermost MBL are needed to verify the proposed mechanisms. Despite these open questions, this study shows that the comparison of measurements at different heights on a research vessel can give new insight into turbulent moisture fluxes during air-sea interaction and may be helpful in the future to estimate the moisture input into the atmosphere from sea spray evaporation.

## 5 Summary and conclusions

In this study, we compared three time series of SWI measurements in water vapour derived from laser spectrometric measurements onboard the RV *Akademik Tryoshnikov* during the Antarctic Circumnavigation Expedition from November 2016 to April 2017. The time series were calibrated and post-processed following a protocol similar to Aemisegger et al. (2012) and Steen-Larsen et al. (2014), as described in Section 2.

These unique five-month time series cover a variety of MBL conditions in different synoptic weather situations and for different small-scale mixing states across the Atlantic and Southern Ocean. We analysed the meridional variations of SWIs in water vapour and their link to meteorological parameters. Overall, the SWI composition in water vapour from the tropics to polar regions reveal distributions similar to the ones known from precipitation measurements at different latitudes. On average, a gradual depletion of heavy isotopes from the tropics to polar regions can be observed, following the evolution of decreasing temperature and specific humidity. The climatic conditions at the measurement and moisture source sites are reflected in the median meridional SWI distribution. The synoptic-timescale variability of SWI signals (interquartile range of meridionally





binned hourly measurements) is highest in extratropical and polar regions. Results from a Lagrangian moisture source analysis reveal that the range of SWI compositions at a specific latitude is strongly linked to the variability in moisture source location and conditions. The MBL water vapour in tropical and subtropical regions has narrow, well-defined moisture source regions aligned with the trade winds. The water vapour sampled in these regions typically originates from the progressive moistening of

5 subsiding mid-tropospheric air masses within anticyclones. In contrast, moisture sources in the extratropics are highly variable as a result of the strong meridional moisture transport typical for the extratropics and generally associated with extratropical cyclones. Cyclone passages lead to alternating moisture transport pathways with equatorward sources in the warm sector and poleward sources in the cold sector. Furthermore, moisture loss during transport, which affects the SWI composition of water vapour, is more important in the extratropics than in subtropical and tropical regions. The range of hourly SWI $\delta-$values in the

10 extratropics under the influence of cyclone passages is larger by an order of magnitude compared to the subtropics and tropics during persistent weather situations. Whether this extratropical SWI variability in the MBL is mainly due to the advection of air masses with different moisture sources and transport characteristics or rather the result of local air-sea fluxes induced by the large-scale advection of air masses is a question that we will address in a future study based on the ACE dataset. Note that we did not encounter tropical deep convective systems in the equatorial Atlantic Ocean during this cruise and, thus, the SWI

variability in the equatorial Atlantic Ocean might be lower during ACE than expected during more convective conditions.

We conducted a thorough quality assessment of the SWI time series, which revealed that the differences between the time series observed at different heights above the ocean is larger than any uncertainty introduced by variations in the instruments' cavity properties, exhaust influence or the calibration procedure. The mean vertical gradients [with 65% percentile range] found for the extratropics are $-0.5\,[-0.9\ldots-0.0]\,\text{‰}\,\text{m}^{-1}$ for $\delta^2\text{H}$, $-0.10\,[-0.16\ldots-0.02]\,\text{‰}\,\text{m}^{-1}$ for $\delta^{18}\text{O}$, and $0.3\,[0.1\ldots0.6]\,\text{‰}\,\text{m}^{-1}$

for $d$.

The vertical SWI differences depend on surface wind speed with larger differences for very high and very low wind speeds compared to intermediate wind speeds. This wind speed dependency is qualitatively interpreted in a framework of different influencing factors, including vertical moisture diffusion, vertical turbulent mixing and sea spray evaporation. Low wind speeds are generally associated with high wave age and low concentrations of sea spray. Therefore, the tendency for larger SWI differ-

25 ences between 8 and 13.5 m a.s.l. at low wind speed can be interpreted as a gradient due to weak vertical mixing of moisture. The effect of non-equilibrium fractionation on the vertical SWI variations is strong in such situations. The small vertical SWI differences at intermediate wind speeds are associated with a lower wave age and might be due to stronger turbulent mixing which leads to a more homogeneous SWI distribution in the lowermost MBL. The large vertical SWI differences at high wind speeds are most likely due to a rough sea, breaking waves (low wave age) and an increased enrichment of the lowermost layers

from water evaporation of sea spray droplets.

The study of vertical SWI variations in MBL water vapour shows the potential of SWIs as tracers for vertical mixing in the lowermost MBL and as an indicator of atmospheric moisture input by sea spray evaporation. For an in-depth understanding and verification of the proposed mechanisms leading to the observed vertical SWI variations, SWI profiles at higher vertical resolution than the two point measurements in this study should be conducted in future studies. Such a setup could provide a

35 framework to better quantify the contribution of sea spray evaporation to MBL moisture.



Overall the presented measurements from the Atlantic and Southern Ocean highlight the large variety of processes at different scales that shape the short-term variability of SWI signals. The interaction of large-scale atmospheric flow features at different latitudes with small-scale turbulent and convective mixing processes is a topic on which SWI measurements from ACE can provide helpful insights in future research.

*Code and data availability.* The ACE datasets are published on the research data repository zenodo: https://zenodo.org/communities/spi-ace/. The SWI-13 datasets are accessible using the following dois: 10.5281/zenodo.3250692 (raw data), XXX (link to zenodo is following, calibrated data) and XXX (link to zenodo is following, calibration scripts). The meteorological data from ACE is published with the doi 10.5281/zenodo.3379590. For access to the SWI-8 dataset, contact Anna Kozachek [kozachek@aari.ru]. The wave data set is published

using the repository of the Australian Antarctic Division (doi: 10.4225/15/5a17923429cb7).

*Author contributions.* IT, AK, PG and YW performed the SWI measurements during ACE. DB, AK and CSL initiated the project leading to the measurements SWI-8 and HW, FA, SP, HS and PG initiated the project leading to the measurements SWI-13. JS provided atmospheric chemistry measurements, SL corrected wind measurements, and AT ocean wave measurements from ACE. IT and FA evaluated the measurements and wrote the paper. All co-authors provided feedback to the manuscript prior to submission.

*Competing interests.* The authors declare that they have no conflict of interest.

*Acknowledgements.* ACE was a scientific expedition carried out under the auspices of the Swiss Polar Institute, supported by funding from the ACE Foundation and Ferring Pharmaceuticals. IT and JS received funding from the ACE Foundation and Ferring Pharmaceuticals. SL received funding from the Swiss Data Science Center. We thank Christian Büchler for the discussions on SWIs measurements in the Atlantic Ocean in the framework of his Bachelor Thesis. MeteoSwiss is gratefully acknowledged for providing access to operational ECMWF analysis

data. HS and YW acknowledge the RCN project FARLAB.



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





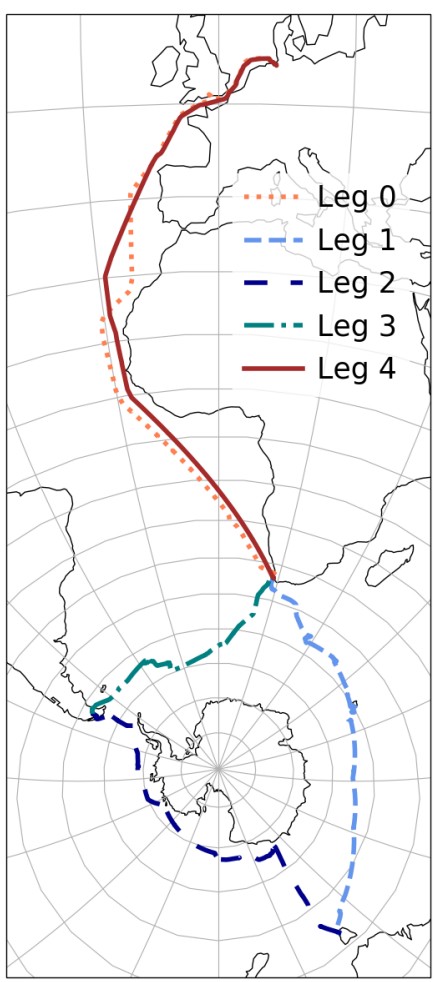

**Figure 1.** Overview of the ACE cruise from November 2016 - April 2017. The five legs (coloured lines) took place from 21 Nov - 15 Dec 2016 (Leg 0), 21 Dec 2017 - 18 Jan 2017 (Leg 1), 22 Jan - 22 Feb 2017 (Leg 2), 26 Feb - 19 Mar 2017 (Leg 3) and 22 Mar - 11 Apr 2017 (Leg 4).


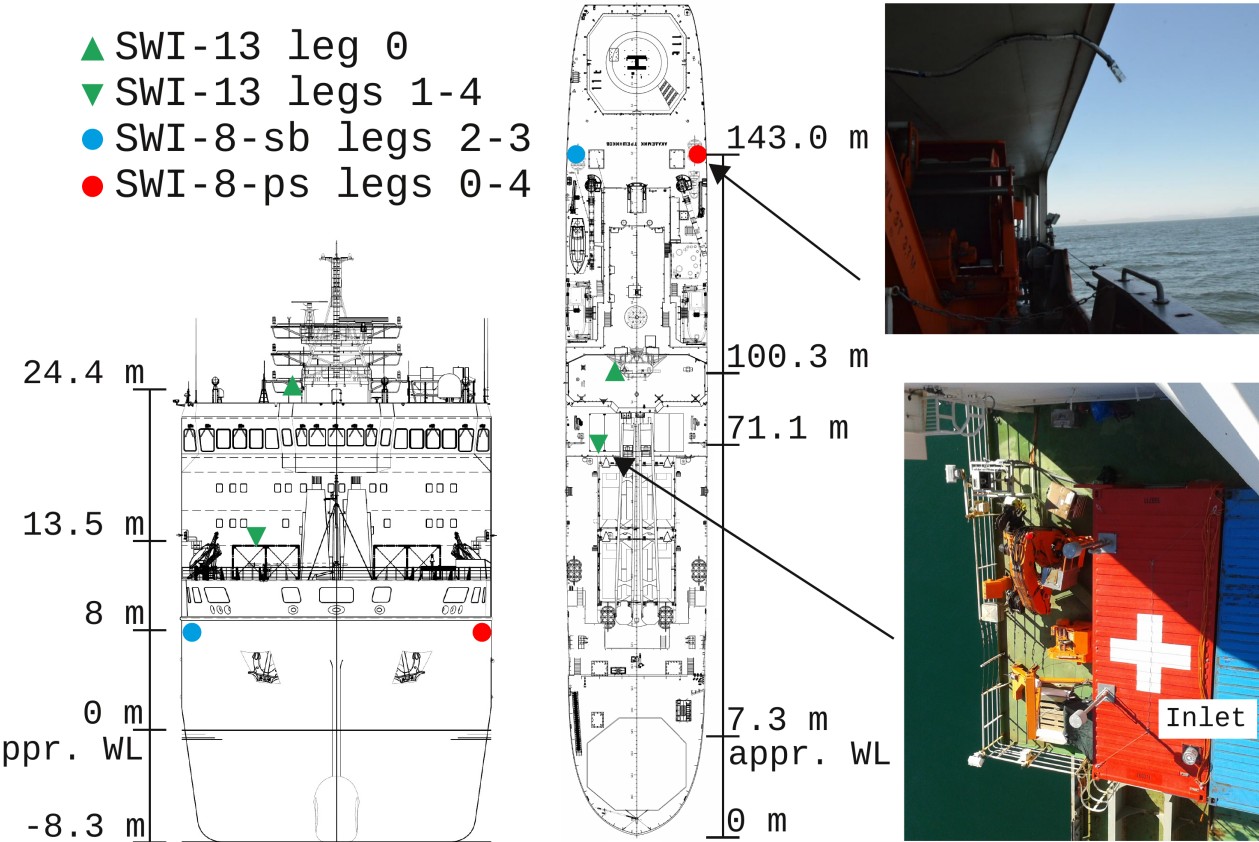

**Figure 2.** Inlet positions on RV *Akademik Tryoshnikov* (adjusted vessel plans from the Arctic and Antarctic Research Institute). Distances are given relative to the approximate water line (appr. WL, [m]) for the front view and relative to the front for the top view, respectively. Pictures of the mounted inlets for SWI-8-ps (top) and SWI-13 legs 1-4 (bottom) are shown to the right.



**Figure 3.** Hourly time series of $\delta^{18}O$, $\delta^2H$, $d$, and $w$ legs 0 and 4 from SWI-13 versus latitude along the ship track. The errorbars denote hourly standard deviations of the 1s data.

**Figure 4.** Time series of hourly $\delta^{18}O$, $\delta^2H$, $d$, and $w$ for legs 1-3 from SWI-13. The errorbars denote hourly standard deviations of the 1s data.



**Figure 5.** Time series of hourly means of **(a)** $\delta^{18}$O, **(b)** $\delta^2$H, **(c)** $d$, and **(d)** $\Delta_{13-8}$ for leg 2 from SWI-13, SWI-8-ps, and SWI-8-sb measurements. $\Delta_{13-8}$ is the difference between SWI-13 and SWI-8-ps. The errorbars in panel **(d)** denote hourly standard deviations of the 1s data.

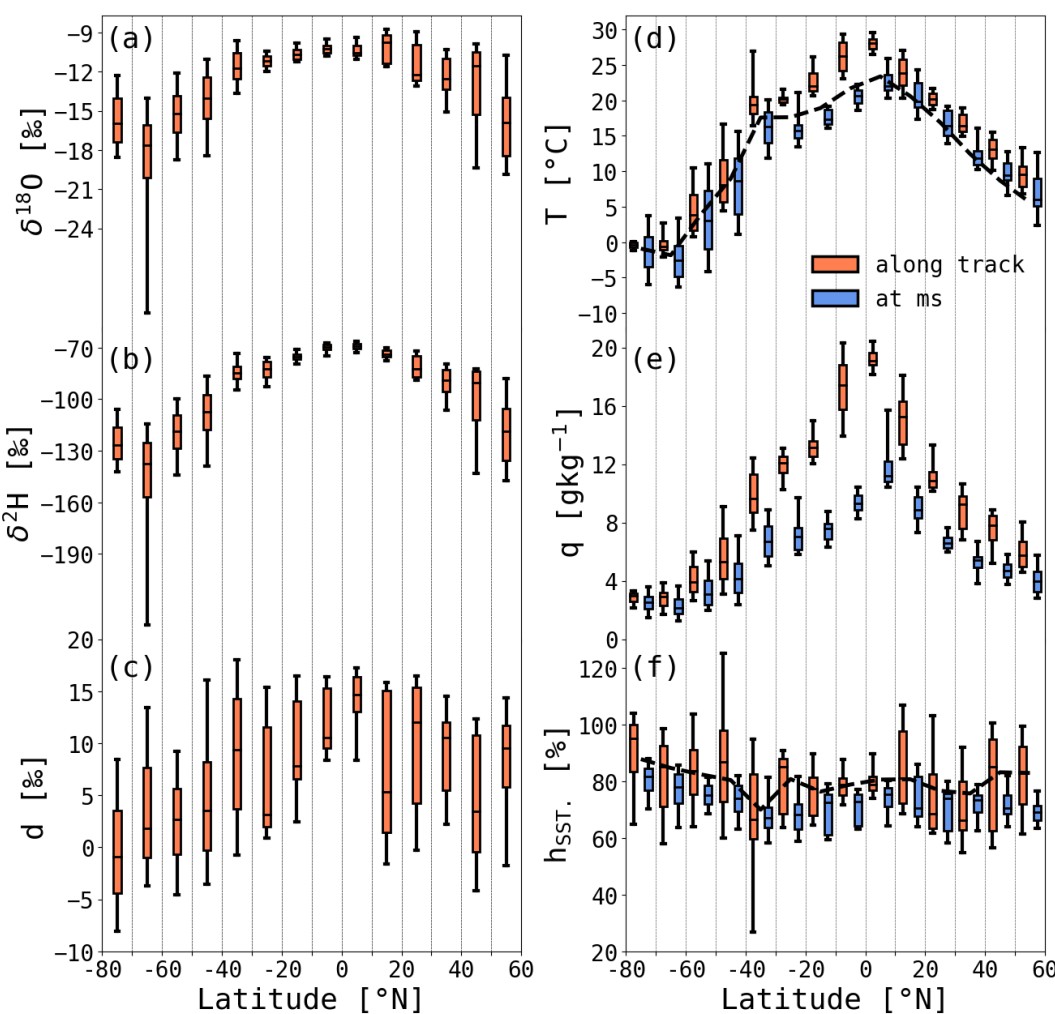

**Figure 6.** Box plots of meridional variations in SWIs and environmental variables for all legs showing mean (black horizontal line in box), interquartile range (boxes) and [5,95]-percentile range (whiskers) of variables in bins of 10° latitudinal width. **(a)** $\delta^{18}$O, **(b)** $\delta^2$H, **(c)** $d$, **(d)** air temperature ($T$), **(e)** specific humidity ($q$), and **(f)** the relative humidity with respect to sea surface temperature ($h_{\mathrm{SST}}$) at the measurement site are shown. Additionally, for $T$, $q$ and $h_{\mathrm{SST}}$ the weighted mean at the moisture sources (ms) is shown **(d-f)**. The black, dashed lines show sea surface temperature from operational ECMWF analysis data **(d)** and the relative humidity at the measurement site **(f)**.



**Figure 7.** Scatterplot of $d$ vs. $h_{\mathrm{SST}}$ coloured by SST. The linear fit to all points with $h_{\mathrm{SST}} <100\%$ (black line) has a Pearson correlation coefficient of -0.73 with the following fitting function: $d(h_{\mathrm{SST}}) = -0.4\text{‰}\%^{-1}{\cdot}h_{\mathrm{SST}}+36.8\text{‰}$. The dashed line shows the linear relationship between $d$ and $h_{\mathrm{SST}}$ from Pfahl and Sodemann (2014). Two time periods with low $h_{\mathrm{SST}}$ are marked: 21 - 25 Dec 2016 (x) and 13 -16 Feb 2017 (+).



**Figure 8.** Contour plots of hourly 75% moisture source regions of water vapour along the ship track for legs 0 - 4 **(a-e)** coloured by time. The colours assign the source regions to the corresponding water vapour along the ship track (black framed line).

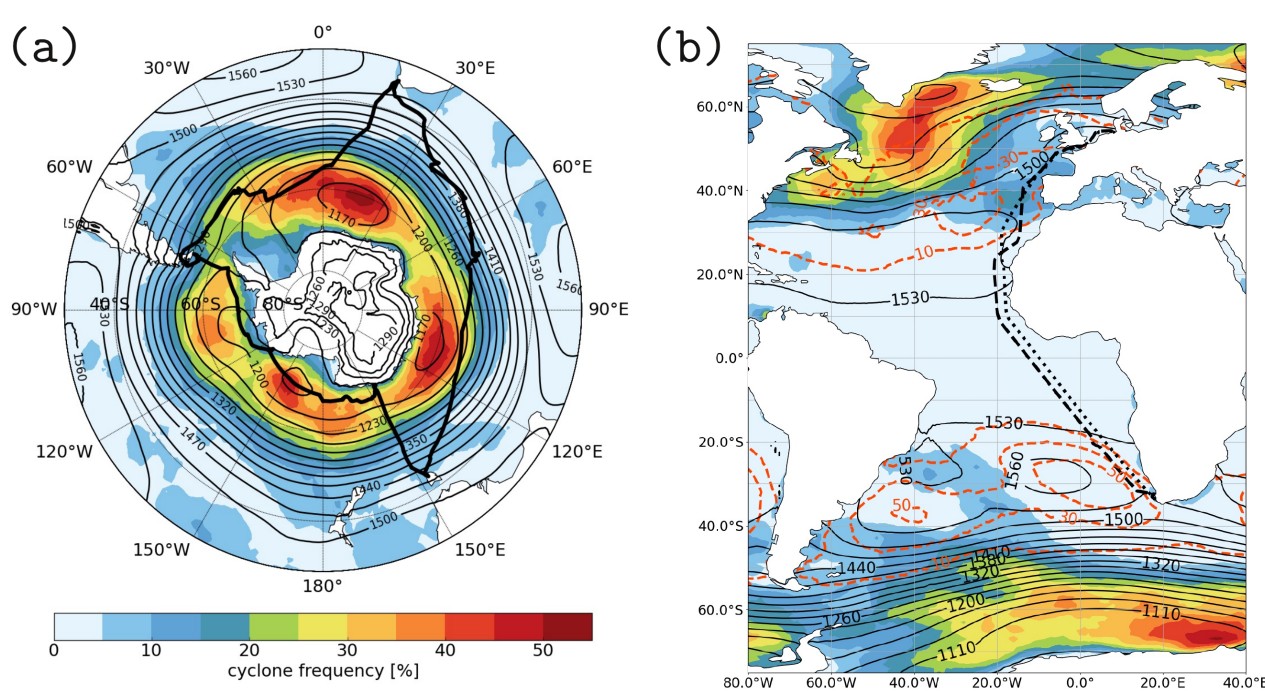

**Figure 9.** Mean cyclone frequencies (coloured contours, [%]) with geopotential height at 850 hPa (black contours, [m]) are shown for legs 1-3 **(a)** and legs 0 and 4 **(b)**. The mean anticyclone frequencies (orange, dashed contours, [%]) for legs 0 and 4 are additionally plotted in panel **(b)**. The thick black line shows the ship tracks for legs 1-3. The dashed and dotted black lines show the ship track of legs 0 and 4, respectively. ECMWF operational analysis data was used to produce this Figure.



**Figure 10.** Scatter plots of the vertical differences between SWI-13 and SWI-8-ps for **(a,d)** $\delta^{18}$O ($\Delta_{13-8}\delta^{18}$), **(b,e)** $\delta^2$H ($\Delta_{13-8}\delta^2$H) and **(c,f)** $d$ ($\Delta_{13-8}d$) versus wind speed for legs 1-3. The colours show the sea spray proxy **(a-c)** or wave age **(d-f)**. The black line represents the mean and standard deviations of the vertical differences in $2\,\mathrm{m\,s^{-1}}$-bins. The vertical lines at the bottom indicate the $2\,\mathrm{m\,s^{-1}}$-bins labelled by the number of measurement points per bin. The labels (I, II, III) at the top correspond to the wind turbulence regimes (see text and Fig. 11). Less points are shown in **(d-f)** than in **(a-c)**, because less wave age than sea spray data is available. The data is shown in 5 min resolution and only every third timestep is plotted in the scatter plot.

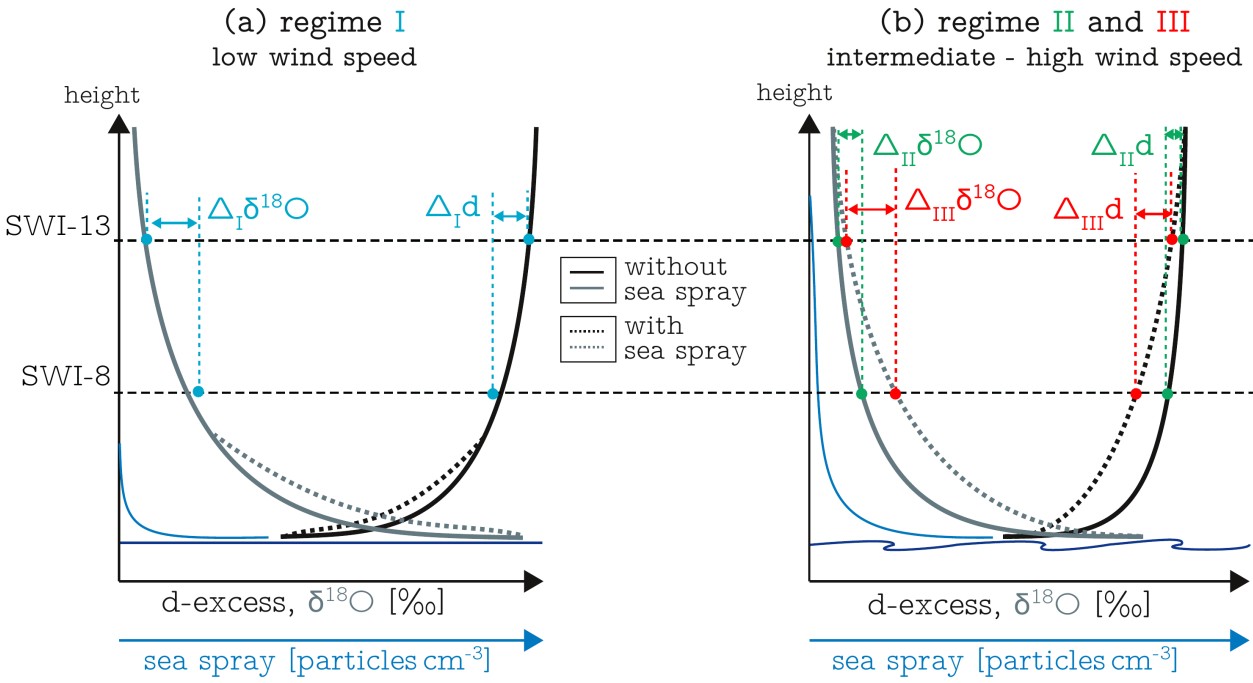

**Figure 11.** Schematic showing the vertical SWI gradient under low **(a)** and intermediate to high **(b)** wind speed conditions illustrating the three boundary layer turbulence regimes (I,II,III). $\Delta_{\mathrm{I}}$, $\Delta_{\mathrm{II}}$ and $\Delta_{\mathrm{III}}$ represent the vertical differences between SWI-13 and SWI-8 for the three regimes. For details see text.





**Table A1.** Calibration versions of SWI-13 and SWI-8-ps used in this study. The following isotope-humidity dependency correction functions are used in the calibration versions: $\mathcal{H}_c$ applies a constant factor of 0 as correction term (i.e. no isotope-humidity dependency correction is used). $\mathcal{H}_{1,min}$ and $\mathcal{H}_{1,max}$ are the best fit correction curves to the SWI-13 calibration runs -/+ 1 standard deviation to estimate the uncertainty of the best fit ($\mathcal{H}_1$) to the calibration runs. Accordingly, $\mathcal{H}_{3,min}$, $\mathcal{H}_{3,max}$ and $\mathcal{H}_3$ are defined for SWI-8. $\mathcal{H}_2$ is the correction curve from Sodemann et al. (2017). The running mean/average column specifies the handling of the times in between calibration runs: *run* refers to 10-day running means used for the calibration runs. For *ave*, the calibration runs of each standard are averaged for each legs and this average value is used for the calibration of the corresponding leg. Version 1 for each dataset (SWI-13 and SWI-8-ps) is the final version used in Section 4.

| Version | isotope-humidity correction | running mean / average |
|---------|------------------------------|-------------------------|
| | SWI-13 | |
| 1 | $\mathcal{H}_1$ | run |
| 2 | $\mathcal{H}_c$ | run |
| 3 | $\mathcal{H}_{1,min}$ | run |
| 4 | $\mathcal{H}_{1,max}$ | run |
| 5 | $\mathcal{H}_1$ | ave |
| 6 | $\mathcal{H}_2$ | run |
| | SWI-8-ps | |
| 1 | $\mathcal{H}_3$ | run |
| 2 | $\mathcal{H}_c$ | run |
| 3 | $\mathcal{H}_{3,min}$ | run |
| 4 | $\mathcal{H}_{3,max}$ | run |
| 5 | $\mathcal{H}_3$ | ave |

**Table A2.** Vertical SWI gradients for the three wind regimes: [I] low wind speed $< 6\,\mathrm{m\,s^{-1}}$, [II] intermediate wind speed between $6\,\mathrm{m\,s^{-1}}$ and $16\,\mathrm{m\,s^{-1}}$, [III] high wind speed $> 16\,\mathrm{m\,s^{-1}}$. The gradients are given in $‰\,\mathrm{m^{-1}}$. In brackets, the 65 % percentile ranges are noted.

| Regime | $\Delta_{13-8}\delta^2\mathrm{H}$ | $\Delta_{13-8}\delta^{18}\mathrm{O}$ | $\Delta_{13-8}d$ |
|--------|-----------------------------------|--------------------------------------|-------------------|
| I-III | $-0.5\,[-0.9\ldots-0.0]$ | $-0.10\,[-0.16\ldots-0.02]$ | $0.3\,[0.1\ldots0.6]$ |
| I | $-0.6\,[-1.0\ldots-0.2]$ | $-0.12\,[-0.18\ldots-0.06]$ | $0.4\,[0.2\ldots0.6]$ |
| II | $-0.4\,[-0.8\ldots-0.0]$ | $-0.09\,[-0.15\ldots-0.01]$ | $0.3\,[0.1\ldots0.5]$ |
| III | $-1.0\,[-1.5\ldots-0.2]$ | $-0.16\,[-0.26\ldots-0.03]$ | $0.3\,[0.0\ldots0.6]$ |

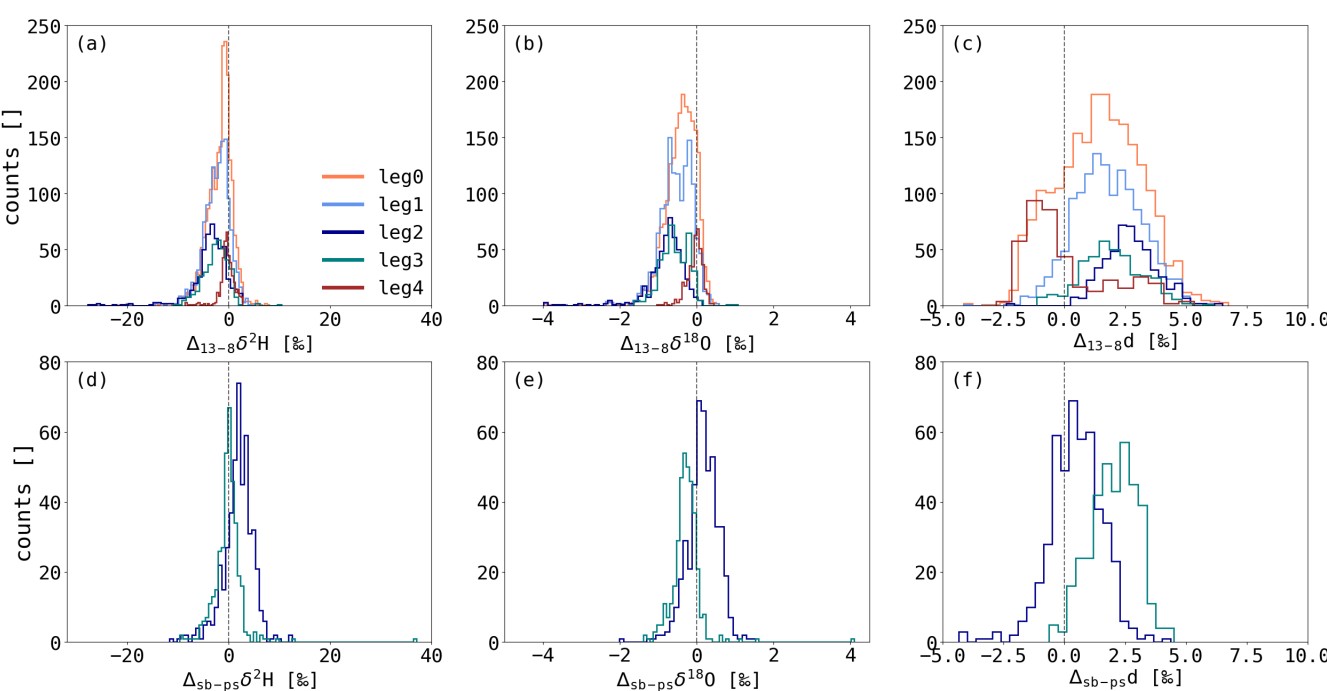

**Figure A1.** Histograms of difference between SWI-13 and SWI-8-ps (**a-c**) and SWI-8-sb and SWI-8-ps (**d-f**) for $\delta^2$H, $\delta^{18}$O and $d$. The histograms are coloured by legs. SWI-8-sb is only available for legs 2 and 3.



**Figure A2.** Normalized histograms showing distribution of difference between SWI-13 and SWI-8-ps ($\Delta_{13-8}$) for $\delta^{18}$O **(a)**, $\delta^2$H **(b)**, and $d$ **(c)** and distribution of absolute wind direction **(d)** for periods without [black, dashed line] and with [grey, solid line] exhaust influence. Westerly wind direction is marked in **(d)** with a thin black line at 270°.

**Figure A3.** Meridional variations of uptake-to-loss ratio (blue circles, []), cumulative loss (orange squares, [g kg$^{-1}$]) and cumulative uptake (green diamonds, [g kg$^{-1}$]) over the 5 days prior to arrival at the measurement sites along backward trajectories. On the x-axis, the mean latitude of the 10° bins is shown.