# Peer review of "Meridional and vertical variations of the water vapour isotopic composition in the marine boundary layer over the Atlantic and Southern Ocean"

_Atmospheric Chemistry and Physics, 2019_

## Referee Comment (RC1) · Anonymous Referee #1 · 20 Oct 2019

This article presents new isotopic measurements during a cruise in the Atlantic Ocean. This cruise has the advantage of spanning a wide latitudinal range. This paper thus contributes to a better description of the latitudinal distribution of isotopic variables and of its variability. Another significant contribution of this paper is to document the isotopic composition at two different altitudes, thus giving some idea of the vertical gradient in the lower part of the boundary layer.

The paper is well written and the figures are of good quality. The measurement methods and possible errors are explained in detail.

[Figure]

In addition to a few minor comments, I have one major comment: I think that the interpretation of the vertical isotopic gradient is faulted because a well-established property of the boundary layer is ignored: molecular diffusion in the boundary layer is not significant beyond the first centimeters above the surface. Fortunately, this fault does not affect the main conclusions of the paper and can be easily corrected.

**1 Major comment: molecular diffusion in the boundary layer is not significant beyond the first centimeters above the surface**

Kinetic fractionation can happen because different isotopologues have different molecular diffusivities (Merlivat (1978); Merlivat and Jouzel (1979); Jouzel and Merlivat (1984)). This plays a role only where molecular diffusion is significant. In the planetary boundary layer, molecular diffusion plays a significant role only in the first millimeters above the surface (Holton (1973); Stull (1988)). Beyond the first centimeters, it can be completely neglected and turbulence is the main factor (see for example fig 7.1 in Stull (1988)). This is a well-established property of the boundary layer. No isotopic fractionation is associated with turbulence. Therefore, molecular diffusion and associated isotopic fractionation cannot explain the vertical gradients in isotopic composition observed at the scale of several meters.

The authors cite Tanny and Cohen (2008) to justify the existence of diffusion processes and associated isotopic fractionation in the turbulent part of the boundary layer. This article is mis-interpreted and mis-used here. Tanny and Cohen (2008) argues that some coherent structures, and not just random eddies, are responsible for the transport in the first centimeters above the surface. This has implications for how we calculate the evaporation flux and its composition (Craig and Gordon (1965) equation). But this article never refutes the well-established property that turbulence dominates beyond the first centimeters above the surface, and never argues that we should modify how

we interpret the vertical gradients of constituents at the scale of several meters. In fact, the processes discussed in Tanny and Cohen (2008) are so close to the surface that they are all supposed to be encapsulated in the evaporation flux equation and in the Craig and Gordon (1965) equation.

The decrease of $\delta D$ and $\delta^{18}O$ with height, at scales of several several meters or larger, has already widely been documented in observations (e.g. Ehhalt (1974); Bailey et al. (2013); Sodemann et al. (2017); Salmon et al. (2019)). There is a simple reason for this negative vertical gradient. In the free troposphere, condensation depletes the water vapor. In the boundary layer, all the vapor ultimately comes from the mixing between freshly evaporated water vapor (enriched), and vapor from the free troposphere that has already undergone condensation (depleted). The proportion of freshly evaporated water vapor decreases with height. This is why $\delta D$ and $\delta^{18}O$ decrease with height in the boundary layer. The fact that numerical models at various resolutions (general circulation models, large-eddy simulations...) can reproduce the negative vertical gradient without representing molecular diffusion within the boundary layer is an additional proof that no molecular diffusion needs to be involved to explain this feature.

The same rationale can apply for d-excess. At the scale of the troposphere, observations suggest that free tropospheric vapor that has undergone significant distillation has a high d-excess (Sayres et al. (2010); Samuels-Crow et al. (2014)). Numerical models at various resolutions can reproduce this increase in d-excess with altitude through the troposphere without representing molecular diffusion within the boundary layer (Bony et al. (2008); Blossey et al. (2010)). Again, this is an additional proof that we can explain an increase in d-excess with altitude without involving molecular diffusion. At the scale of the boundary layer, vertical profiles can be more diverse (e.g. Sodemann et al. (2017); Salmon et al. (2019)), maybe due to the non-monotonic evolution along the distillation and to the stronger effect of droplet or rain evaporation (Salmon et al. (2019)). But in the case where d-excess in the free troposphere or in the upper part of the boundary layer is high, following the same rationale as for $\delta^{18}O$ and $\delta D$, we expect

d-excess to increase with height in the boundary layer.

As a consequence, several sentences need to be modified:

- p 2 l 5: "and dominating effect from non-equilibrium fractionation" needs to be removed. Non-equilibrium fractionation where? In the atmosphere or at the surface? If you mean in the atmosphere, it's wrong as explained above. If you mean at the surface, it's probably wrong as well, because weak vertical turbulent mixing is probably associated with higher relative humidity at the surface and thus weaker kinetic fractionation.

- p 3 l 31- p 4 l 4: to be removed, because Tanny and Cohen (2008) is misinterpreted here.

- p 16 l 21-22: to be removed.

- p 17 l 27-30: this hypothetically turbulence-free atmosphere is too unrealistic to be used as a started point for interpreting vertical gradients at the scale of several meters. Non-equilibrium fractionation only impacts the composition of the evaporation flux, not the vertical gradients. The 1st process may thus be either removed, or replaced by the fact that the isotopic composition of the evaporation flux is more enriched and has a lower d-excess than the water vapor that has already undergone distillation.

- p 18 l 2-4: remove "Due to moisture ... weak turbulence". The previous sentence is a sufficient explanation and can be replaced by: "This leads to ... with strong vertical moisture and isotopic gradients".

- p 18 l 7: remove "weakens ... by diffusion"

- p 19 l 23: remove "vertical moisture diffusion, "

- p 19 l 26: remove "The effect ... such situations".

**2  Minor comments**

- p 1 l 16: "moisture loss during transport": if moisture loss happens in clouds, it does not directly impact the isotopic composition of the vapor near the surface. Replace by something like: "mixing with air masses that have lost moisture during their transport"?

- p 1 l 18: "low" -> "weak".

- p 1 l 19: "at different heights": this misleads the reader to think there are many heights -> specify "at 8m and 13m".

- p 2 l 12-13: These citations are not very relevant here. Cite papers that really show the point. These cited papers just look at the effect of atmospheric processes on the isotopic composition, but we knew about these atmospheric processes long before 2018. These papers can be usefully cited elsewhere.

- p3 l 8: "MBL moisture budget": we do not need isotopes for this -> "MBL isotopic budget"

- p 3 l 30: how do you define the top of the MBL here? Is it the cloud top or cloud base?

- p 4 l 8: here a bit more background would be necessary to understand what is the wave age. Is it related to a time unit, e.g. seconds or minutes or hours? What does it physically mean when it is <1?

- p11 l 24: "to assess moisture sources": we do not need SWI for this -> "to assess the effect on SWI of moisture sources"

- p 14 l 21: "not shown": it would be useful to add this plot in the main text or in SI.

- p 15 l 5: remove "(Fig 8)" which does not show the point.

- p 15 l 20: "larger degree of precipitation" -> "larger variability in the precipitation": here you want to explain the isotopic variability, not its mean value.

- p 18 l 17-19: to verify these mechanisms, observations won't be sufficient. Large-eddy simulations would be useful. A sentence to mention this could be added.

- p 19 l 9: "more important" -> "more variable" (same comment as p 15 l 20)

**References**

Bailey, A., Toohey, D., and Noone, D. (2013). Characterizing moisture exchange between the hawaiian convective boundary layer and free troposphere using stable isotopes in water. *Journal of Geophysical Research: Atmospheres*, 118(15):8208–8221.

Blossey, P. N., Kuang, Z., and Romps, D. M. (2010). Isotopic composition of water in the tropical tropopause layer in cloud-resolving simulations of an idealized tropical circulation. *J. Geophys. Res.*, 115:D24309, doi:10.1029/2010JD014554.

Bony, S., Risi, C., and Vimeux, F. (2008). Influence of convective processes on the isotopic composition (deltaO18 and deltaD) of precipitation and water vapor in the Tropics. Part 1: Radiative-convective equilibrium and TOGA-COARE simulations. *J. Geophys. Res.*, 113:D19305, doi:10.1029/2008JD009942.

Craig, H. and Gordon, L. I. (1965). Deuterium and oxygen-18 variations in the ocean and marine atmosphere. *Stable Isotope in Oceanographic Studies and Paleotemperatures*, Laboratorio di Geologia Nucleate, Pisa, Italy:9–130.

Ehhalt, D. H. (1974). Vertical profiles of hto, hdo, and h2o in the troposphere. *NCAR technical note*, NCAR-TN-STR-100.

Holton, J. R. (1973). An introduction to dynamic meteorology. *American Journal of Physics*, 41(5):752–754.

Jouzel, J. and Merlivat, L. (1984). Deuterium and oxygen 18 in precipitation: modeling of the isotopic effects during snow formation. *J. Geophys. Res.*, 89:11:749.

Merlivat, L. (1978). Molecular diffusivities of H2O16, HDO16, and H2O18 in gases. *J.Chem. Phys.*, 69:2864–2871.

Merlivat, L. and Jouzel, J. (1979). Global climatic interpretation of the Deuterium-Oxygen 18 relationship for precipitation. *J. Geophys. Res.*, 84:5029–5332.

Salmon, O. E., Welp, L. R., Baldwin, M. E., Hajny, K. D., Stirm, B. H., and Shepson, P. B. (2019). Vertical profile observations of water vapor deuterium excess in the lower troposphere. *Atmospheric Chemistry and Physics*, 19(17):11525–11543.

Samuels-Crow, K. E., Galewsky, J., Sharp, Z. D., and Dennis, K. J. (2014). Deuterium excess in subtropical free troposphere water vapor: Continuous measurements from the chajnantor plateau, northern chile. *Geophysical Research Letters*, 41(23):8652–8659.

Sayres, D. S., Pfister, L., Hanisco, T. F., Moyer, E. J., Smith, J. B., Clair, J. M. S., O'Brien, A. S., Witinski, M. F., Legg, M., and Anderson, J. G. (2010). Influence of convection on the water isotopic composition of the tropical tropopause layer and tropical stratosphere,. *J. Geophys. Res.*, 11:D00J20, doi:10.1029/2009JD013100.

Sodemann, H., Aemisegger, F., Pfahl, S., Bitter, M., Corsmeier, U., Feuerle, T., Graf, P., Hankers, R., Hsiao, G., Schulz, H., et al. (2017). The stable isotopic composition of water vapour above corsica during the hymex sop1 campaign: insight into vertical mixing processes from lower-tropospheric survey flights. *Atmospheric Chemistry and Physics*, 17(9):6125–6151.

Stull, R. B. (1988). *An intruduction to boundary layer meteorology*. Dordrect Kluwer.

Tanny, J. and Cohen, J. (2008). Revisiting the boundary layer structure used in craig and gordon's model of isotope fractionation in evaporation. *Isotopes in environmental and health studies*, 44(1):11–21.
* * *

---

## Referee Comment (RC2) · Anonymous Referee #2 · 26 Oct 2019

Review of Âń Meridional and vertical variations of the water vapour isotopic composition in the marine boundary layer over the Atlantic and Southern OceanÂż by Iris Thurnherr et al,

The manuscript presents a time series of vapour isotopic composition from the marine boundary layer from an expedition that realised a circumpolar around Antarctica. The dataset presented in this manuscript is the first of its kind: providing an unique spatial coverage of the vapour isotopic composition of the Southern Ocean, at two different heights in the marine boundary layer. The quality of the produced data is rigorously

assessed by a large number of calibrations, tests of the impact of the ship itself on the measurements and the presence of multiple instruments. The manuscript focuses on two aspects of the results: first, the meridional variations of the isotopic composition in the marine boundary layer, and second, the vertical variations. The authors have developed a physical qualitative framework to explain the results. While the analysis of the result is done thoroughly and with adequate justification, no attempt to use previous theory of the formation of the isotopic composition of the vapour in the marine boundary layer is presented here. A large amount of the theories were set in a period where analytical capabilities did not provide such extensive dataset, and it is an important duty that to confront these theories to field measurements. I believe these changes will be relatively easy for the authors, and that they will strengthen an already important manuscript for the link between isotopic composition and marine boundary layer dynamics.

Main comments: 1. Uncertainty evaluation: The second message of the manuscript (section 4.2) details the vertical differences between two infrared spectrometers that were installed at 8 and 13 meters, respectively. On average, significant differences are observed between these two instruments. A significant amount of work is dedicated in this manuscript into characterising the instruments performances. Yet, the results in section 4.2 do not include an error bar for which the differences are significant between the two instruments. In particular, in Fig. 10, a significant number of datapoints presented have very small difference (< 0.2 ‰ in $\delta$18O for instance). Considering the precisions of the instruments (in particular the 2120), it is difficult to assess the relevance of these datapoints. This is a key aspect to be able to justify the wind speed dependency, and it seems that most of the results necessary to evaluate the statistical significance of the results are already presented here. I would suggest make use of the standard deviation of the differences (for instance in Fig. 5) and use pertinent statistical tests (for instance Kruskall Wallis tests) to evaluate in which cases are the differences statistically significant.

2. In the manuscript, the authors do not provide any quantitative evaluation of the vertical differences of the isotopic composition in the marine boundary layer. Yet, formulations have been predicted, based on very limited number of observations compared to this study. While I generally agree with the qualitative proposition of the authors, I believe that they should have tested previous formulations. From articles already mentioned in the manuscript, I would suggest to compare their results to models of isotopes in the boundary layers, namely Craig (1965), Merlivat (1978), or again Benetti et al. (2018). I would suggest to use formulations developed in Cappa et al. (2003), and the parametrisations of Merlivat (1978) for the dependency of the diffusion with turbulence. I suggest that these parametrisations, which already include an increasing impact of turbulence with wind speed, should be tested. Due to the considerable amount of data of the authors, I would suggest evaluating this on typical cases (for instance, the regimes [I], [II] and [III] identified by the authors. Also, as here $\delta$18O is expected to decrease monotonously with height, I would suggest that the authors identify the different contributions to d-exc and $\delta$18O (or $\delta$D and $\delta$18O) in an isotope-isotope space (for instance $\delta$D vs $\delta$18O) and illustrate which process is characterised with slopes higher or lower than the meteoric water line.

Minor comments:

Page 2, Line 10: "The atmospheric water cycle is an essential component of the Earth's climate system" The water cycle is not just atmospheric by definition. Page 2, line 24: "SWIs are tracers of moist atmospheric processes because they record phase changes in the atmosphere." What is a moist atmospheric process ? Sentence unclear Page 3, line 28 to 35: I would suggest include articles such as (Craig, 1965;Cappa et al., 2003). Page 4, line 5 to 19: The link with the isotopes and their limits in this context is missing.

Benetti, M., Lacour, J. L., Sveinbjörnsdóttir, A., Aloisi, G., Reverdin, G., Risi, C., Peters, A., and Steen‐Larsen, H.: A framework to study mixing processes in the marine boundary layer using water vapor isotope measurements, Geophysical Research

Letters, 45, 2524-2532, 2018. Cappa, C. D., Hendricks, M. B., DePaolo, D. J., and Cohen, R. C.: Isotopic fractionation of water during evaporation, J. Geophys. Res.-Atmos., 108, 10, 10.1029/2003jd003597, 2003. Craig, H., Gordon, A. : Deuterium and oxygen 18 variations in the ocean and the marine atmosphere, Stable Isotopes in Oceanic Studies and Paleotemperatures, 130, 1965. Merlivat, L.: The dependence of bulk evaporation coefficients on air-water interfacial conditions as determined by the isotopic method, Journal of Geophysical Research: Oceans, 83, 2977-2980, 10.1029/JC083iC06p02977, 1978.

---

## Referee Comment (RC3) · Anonymous Referee #3 · 11 Nov 2019

Review of ACP-2019-782

SUMMARY This paper by Thurnherr et al. is a detailed survey of exploration of water vapor isotopic composition in the marine boundary layer (MBL) in the Atlantic Ocean and Southern Ocean over a wide range of latitudes. The Antarctica Circumnavigation Expedition (ACE), which occurred from Nov 2016 through April 2017, took a five-month continuous time series of stable water isotope (SWI) measurements and analyzed the causes of SWI variations. Specifically, this paper explores in detail the SWI variations in the MBL with latitude, the relation between large-scale circulation and measured SWI

at different latitudes, and the effects of near-surface wind speed and ocean surface state on SWI variations with altitude.

SIGNIFICANCE: Stable water isotopes (SWI) are unique tracers of the atmospheric water cycle, and this is a pioneering survey of SWI in the Atlantic and Southern Ocean, both the first measurements and detailed moisture source analysis. The measurements are of high quality, as demonstrated by extensive calibration. I highly recommend publication after minor technical corrections, in particular minor changes to the figures.

TECHNICAL CORRECTIONS:

1. Page 2, line 24: Unless you think that this is obvious, I recommend the authors add that R(VSMOW2) is multiplied by 2 for the two possible positions of the isotope within the water molecule.

2. Page 2, line 26: Equilibrium fractionation is not the only type of isotopic fractionation. Non-equilibrium fractionation by diffusion is also isotopic fractionation. I recommend the authors modify as follows: 'The difference in saturation vapor pressure between heavy and light isotopologues causes one type of isotopic fractionation . . .'

3. Page 9, line 14: The authors state that horizontal differences between SWI-8-sb ad SWI-8-ps sensors are smaller than vertical differences, but Figure 5 seems to indicate that d has large horizontal differences comparable to vertical differences in d.

4. Page 13, line 23 incorrect figure citation, should be: 'The meridional distribution of d (Fig. 6c) . . .'

5. Page 14, line 17 incorrect figure citation, should be: Fig. 6 a,b,c.

6. Page 15, line 25 incorrect figure citation, should be: Fig. 6f.

7. Figure 1, page 26: I suggest that the authors add latitude labels on Figure 1, it would help with interpretation.

[Figure]

SPELLING/TYPOS:

1. Page 6, line 12 (also lines 19, 20, 33): Is apostrophe 12'000 standard Swiss notation for 12 000? I recommend no apostrophe to avoid confusion.

2. Page 13, line 6: change 'warmer T' to 'higher T'.

3. Page 13, line 12: spelling typo, should be 'meridional'.

4. Page 13, line 33: spelling typo, should be 'Agulhas'.

Figure 4, page 29: add abscissa (x-axis) label: "Date (dd-mm)".

3. Figure 5, page 30: add abscissa (x-axis) label: "Date (dd-mm)".

4. Figure 6, page 31: the letters 6a, 6b, etc. in the text do not appear to match the order of figures in Figures 6. Please recheck.

5. Figure 6 caption, page 31, and Page 18, line 32: I recommend changing 'site' to 'location' because site implies a fixed location whereas you are measuring at many latitudes along the ship track.

6. Figure 10 caption, page 35: in the next-to-last sentence change 'Less points' to 'Fewer points. . .'

---

## Author Comment (AC1) · 19 Jan 2020

Authors' reply to comment on *Meridional and vertical variations of the water vapour isotopic composition in the marine boundary layer over the Atlantic and Southern Ocean* by Anonymous Referee #1

We would like to thank the reviewer for taking the time to read our manuscript and giving detailed feedback, which helped us to improve the manuscript. Please find our answers to the comments (*in italics*) in the following. Citations from the paper are marked in blue, and **blue bold** refers to text added/changed in the revised manuscript.

**1 Major comment: molecular diffusion in the boundary layer is not significant beyond the first centimeters above the surface**

*Kinetic fractionation can happen because different isotopologues have different molecular diffusivities (Merlivat (1978); Merlivat and Jouzel (1979); Jouzel and Merlivat (1984)). This plays a role only where molecular diffusion is significant. In the planetary boundary layer, molecular diffusion plays a significant role only in the first millimeters above the surface (Holton (1973); Stull (1988)). Beyond the first centimeters, it can be completely neglected and turbulence is the main factor (see for example fig 7.1 in Stull (1988)). This is a well-established property of the boundary layer. No isotopic fractionation is associated with turbulence. Therefore, molecular diffusion and associated isotopic fractionation cannot explain the vertical gradients in isotopic composition observed at the scale of several meters.*

The reviewer is correct; non-equilibrium fractionation during ocean evaporation due to diffusion mainly occurs in the first centimeters above the ocean surface. Even though non-equilibrium fractionation is not expected to occur at several meters height above the sea surface, it still affects the isotopic composition of water vapour at higher levels through transport processes and due to turbulent mixing that mixes the freshly evaporated water vapour with water vapour at higher levels. Thus, the *effect* of non-equilibrium fractionation also impacts the vertical isotope profiles in the whole boundary layer.
Although we do see a stronger influence of non-equilibrium fractionation on the vertical isotope profile at low wind speeds (as shown in detail in our reply to referee 2), we agree that the influence of non-equilibrium fractionation alone cannot explain the observed vertical gradient. Therefore, we will not discuss the effects of non-equilibrium fractionation and its influence on the vertical SWI gradients in this study, and we deleted the non-equilibrium fractionation from the framework explaining the observed vertical profiles (see also our response to the following comments).

*The authors cite Tanny and Cohen (2008) to justify the existence of diffusion processes and associated isotopic fractionation in the turbulent part of the boundary layer. This article is mis-interpreted and mis-used here. Tanny and Cohen (2008) argues that some coherent structures, and not just random eddies, are responsible for the transport in the first centimeters above the surface. This has implications for how we calculate the evaporation flux and its composition (Craig and Gordon (1965) equation). But this article never refutes the well-established property that turbulence dominates beyond the first centimeters above the surface, and never argues that we should modify how we interpret the vertical gradients of constituents at the scale of several meters. In fact, the processes discussed in Tanny and Cohen (2008) are so close to the surface that they are all supposed to be encapsulated in the evaporation flux equation and in the Craig and Gordon (1965) equation.*

Thank you for this comment. We removed the reference to Tanny and Cohen (2008), which was indeed misleading in the context of the interpretation of the vertical profiles over 13 m above the ocean surface.

*The decrease of δD and δ¹⁸O with height, at scales of several meters or larger, has already widely been documented in observations (e.g. Ehhalt (1974); Bailey et al. (2013); Sodemann et al. (2017); Salmon et al. (2019)). There is a simple reason for this negative vertical gradient. In the free troposphere, condensation depletes the water vapor. In the boundary layer, all the vapor ultimately comes from the mixing between freshly evaporated water vapor (enriched), and vapor from the free troposphere that has already undergone condensation (depleted). The proportion of freshly evaporated water vapor decreases with height. This is why δD and δ¹⁸O decrease with height in the boundary layer. The fact that numerical models at various resolutions (general circulation models, large-eddy simulations...) can reproduce the negative vertical gradient without representing molecular diffusion within the boundary layer is an additional proof that no molecular diffusion needs to be involved to explain this feature. The same rationale can apply for d-excess. At the scale of the troposphere, observations suggest that free tropospheric vapor that has undergone significant distillation has a high d-excess (Sayres et al. (2010); Samuels-Crow et al. (2014)). Numerical models at various resolutions can reproduce this increase in d-excess with altitude through the troposphere without representing molecular diffusion within the boundary layer (Bony et al. (2008); Blossey et al. (2010)). Again, this is an additional proof that we can explain an increase in d-excess with altitude without involving molecular diffusion. At the scale of the boundary layer, vertical profiles can be more diverse (e.g. Sodemann et al. (2017); Salmon et al. (2019)), maybe due to the non-monotonic evolution along the distillation and to the stronger effect of droplet or rain evaporation (Salmon et al. (2019)). But in the case where d-excess in the free troposphere or in the upper part of the boundary layer is high, following the same rationale as for δ¹⁸O and δD, we expect d-excess to increase with height in the boundary layer.*

Thank you for pointing this out. We agree that diffusive processes are not needed to explain the observed vertical SWI gradients. Therefore, we do not use this process to explain the vertical SWI gradients in this study anymore. We explain the changes made to the manuscript in the following, point by point:

*As a consequence, several sentences need to be modified:*

   a) *p 2 l 5: "and dominating effect from non-equilibrium fractionation" needs to be removed. Non-equilibrium fractionation where? In the atmosphere or at the surface? If you mean in the atmosphere, it's wrong as explained above. If you mean at the surface, it's probably wrong as well, because weak vertical turbulent mixing is probably associated with higher relative humidity at the surface and thus weaker kinetic fractionation.*

   We removed this part of the sentence.
   We would still like to mention in this reply, that non-equilibrium fractionation at the ocean surface can influence the isotopic composition of the MBL by upward turbulent mixing. During ACE, we encountered low wind speeds at high and low relative humidity (see figure A). Weak turbulent mixing can also occur together with low relative humidity and, thus, can also be associated with substantial non-equilibrium fractionation (see blue points at low $RH_{SST}$ in Fig. A).

   We changed the text as follows:

"Using sea spray concentrations and sea state conditions, we show that the vertical SWI gradients are particularly large during high wind speed conditions with increased contribution of sea spray evaporation or during low wind speed conditions due to weak vertical turbulent mixing "

[Figure]

*Figure A: Scatter plot of ACE measurements showing relative humidity with respect to sea surface temperature [RH$_{SST}$] versus the vertical difference in d-excess [$\Delta_{13-8}$ d] coloured by wind speed measured at 30 m above the ocean surface.*

b)  *p 3 l 31- p 4 l 4: to be removed, because Tanny and Cohen (2008) is mis-interpreted here.*

Removed as suggested.

c)  *p 16 l 21-22: to be removed.*

Removed as suggested.

d)  *p 17 l 27-30: this hypothetically turbulence-free atmosphere is too unrealistic to be used as a started point for interpreting vertical gradients at the scale of several meters. Non-equilibrium fractionation only impacts the composition of the evaporation flux, not the vertical gradients. The 1st process may thus be either removed, or replaced by the fact that the isotopic composition of the evaporation flux is more enriched and has a lower d-excess than the water vapor that has already undergone distillation.*

We removed the first process and changed the text to:
**"Two main processes are taken into account in this framework: (1) vertical turbulent mixing, which increases with wind speed, leads to a well-mixed atmospheric layer close to the ocean surface and, thus, weakens the vertical SWI gradients; and (2) the sea state**

**determines the production of sea spray and the influence of sea spray evaporation on SWI composition. The proposed framework considers the three wind regimes introduced in section 4.2.1, in which these two processes are expected to differ in strength. As a consequence, vertical turbulent mixing and sea spray evaporation exert a varying influence on the vertical SWI gradient in the lowermost MBL .”**

e) *p 18 l 2-4: remove "Due to moisture ... weak turbulence". The previous sentence is a sufficient explanation and can be replaced by: "This leads to ... with strong vertical moisture and isotopic gradients".*

We changed the text as follows:

"First, for low wind speed conditions with high wave age in regime I (Fig. 12a), weak vertical turbulent moisture transport is expected. **If the MBL vertical moisture gradient results from linear mixing of freshly evaporated water vapour from the ocean surface with moisture from the free troposphere, which likely experienced condensation previously, then lower layers are expected to have higher δ-values than upper layers. In such a scenario, weak vertical mixing leads to strong gradients of specific humidity and δ-values. Non-equilibrium fractionation at the ocean surface during evaporation strongly impacts $d$ in the ocean evaporation flux. Therefore, the vertical $d$-gradient in the lower MBL depends on the strength of non-equilibrium fractionation at the ocean surface. If we assume a simple "two end-member"-mixing process in the MBL of freshly evaporated water vapour with free tropospheric air masses that have undergone substantial rainout, the vertical gradient in $d$ is defined by the difference in $d$ between these two end-members. Free tropospheric air masses, which have lost a major fraction of their water vapour during rainout, show a $d$ in water vapour which closely follows a Rayleigh distillation process (Samuels-Crow et al., 2014) and are expected to have high $d$ by the definition of $d$ (see e.g. Dütsch et al., 2017). $d$ in freshly evaporated water vapour is therefore expected to remain below $d$ of free tropospheric air masses that have undergone substantial rainout previously. An effect by sea spray evaporation is not expected in this wind regime as only little sea spray is produced at low wind speeds. This simple interpretation framework could explain the observed conditions with enhanced gradients in $\delta^{18}O$, $\delta\,^2H$ and $d$ at low wind speeds (regime I) compared to medium wind speeds (regime II). However, recent studies (e.g. Sodemann et al., 2017 and Salmon et al., 2019) showed that the vertical gradients in particular of $d$ rarely follow a simple two end-member mixing model. Differential transport processes in the boundary layer as well as convective plumes with enriched water vapour (and lower $d$) are probably responsible for the large variability in the observed vertical isotope profiles. Therefore, further analysis, which goes beyond the scope of this study, is needed to quantify the wind dependency of non-equilibrium fractionation and its effect on the vertical $d$ gradient in the MBL."**

f) *p 18 l 7: remove "weakens ... by diffusion"*

Removed as suggested.

g) *p 19 l 23: remove "vertical moisture diffusion, "*

Removed as suggested.

h) *p 19 l 26: remove "The effect ... such situations".*

Removed as suggested.

**2 Minor comments**

a)  *p1 l 16: "moisture loss during transport": if moisture loss happens in clouds, it does not directly impact the isotopic composition of the vapor near the surface. Replace by something like: "mixing with air masses that have lost moisture during their transport"?*

We analysed backward trajectories computed from a vertical stack of starting points within the boundary layer above the ship's position. Figure B illustrates that the changes in specific humidity, on which the moisture loss to source ratio is based, also occur in the air masses arriving close to the ocean surface. There are increases as well as decreases in specific humidity along these trajectories, and the decreases in specific humidity can occur close to the arrival time and mostly during the ascent of air masses which generally means that cloud formation and precipitation changes the isotopic composition of the air masses directly.

Therefore, we do not change this sentence.

[Figure]

*Figure B: 10-day backwards trajectories from legs 1-3 starting between 0 and 40 hPa above sea level pressure. Colours denote changes in specific humidity between two consecutive time steps (positive value denotes increase in specific humidity).*

b)  *p1 l 18: "low" -> "weak".*

Changed as suggested:
"In the subtropics and tropics, persistent anticyclones lead to well-confined narrow easterly moisture source regions, which is reflected in the **weak** SWI variability in these regions."

c)  *p1 l 19: "at different heights": this misleads the reader to think there are many heights -> specify "at 8m and 13m".*

Changed as suggested:

"Furthermore, the ACE SWI time series recorded **at 8 m and 13 m** above the ocean surface provide estimates of vertical SWI gradients in the lowermost marine boundary layer."

d) *p2 l 12-13: These citations are not very relevant here. Cite papers that really show the point. These cited papers just look at the effect of atmospheric processes on the isotopic composition, but we knew about these atmospheric processes long before 2018. These papers can be usefully cited elsewhere.*

Replaced by:

"The main source for atmospheric water in oceanic regions is ocean evaporation which is strongly influenced by the large-scale atmospheric flow **(Simmond and King 2004, Papritz et al., 2014)** as well as small-scale turbulent and convective mixing **(Jabouille et al. 1995, Sherwood et al. 2010)**."

e) *p3 l 8: "MBL moisture budget": we do not need isotopes for this -> "MBL isotopic budget"*

Unchanged: SWI help to pinpoint the influence of different moist processes such as sea spray evaporation or below-cloud effects (e.g. precipitation evaporation), which are difficult to identify without the use SWI (e.g. Aemisegger et al., 2015; this study)

f) *p3 l 30: how do you define the top of the MBL here? Is it the cloud top or cloud base?*

For the three-layer model, we define the MBL top as the cloud base because cloud processes are not included in this simplified model of MBL processes.

g) *p 4 l 8: here a bit more background would be necessary to understand what is the wave age. Is it related to a time unit, e.g. seconds or minutes or hours? What does it physically mean when it is <1?*

The text was changed to:

"**The sea state can be described by the dimensionless wave age, which is the ratio of the phase speed of the dominant wave component of the sea state to wind speed (Young, 1999). The wave age describes the ability of waves to absorb energy from the wind and hence represents stages of their development process. On the one hand, when waves are young (wave age ~<1.0), waves travel slower than wind and thus are strongly forced by the atmosphere. As a result, waves absorb energy from the wind, grow rapidly and eventually break, generating sea spray (see e.g. Toffoli et al., 2017). When the sea state is mature (wave age >1.0), on the other hand, waves travel faster than the wind and no longer absorb energy from it. Under these circumstances, waves are independent from the wind and assume a gently sloping profile, which makes them less prone to breaking and spray generation."**

h) *p11 l 24: "to assess moisture sources": we do not need SWI for this -> "to assess the effect on SWI of moisture sources"*

We changed the text as suggested:
"The five-month time series of SWIs in water vapour provide the unique opportunity to assess **the effect** of moisture source and transport processes **on SWIs** in the MBL on various time scales and under diverse climatic conditions."

i) *p 14 l 21: "not shown": it would be useful to add this plot in the main text or in SI.*

The meridional variability in source latitude is already shown indirectly in Fig. 8. As shown in the figure below, there is an increase in the standard deviation of the weighted mean moisture source latitude ($lat_{ms,sd}$, blue line) in the extratropics compared to the tropics, except for the bin between 0 and 10°N. The ITCZ lies within this bin, with moisture source locations either north or south of the ITCZ. This induces a large $lat_{ms,sd.}$ To keep the paper concise and limit the number of figures, we don't show it in the paper, but add it to the supplementary material.

[Figure]

*Figure C: Box plots of meridional variations in the weighted mean moisture source latitude for all legs showing mean (black horizontal line in box), interquartile range (orange boxes) and [5,95]-percentile range (whiskers) in bins of 10° latitudinal width. The blue line shows the standard deviation of the weighted mean moisture source latitudes of each 10°-bin.*

j) *p15 l 5: remove "(Fig 8)" which does not show the point.*

Removed as suggested.

k) *p 15 l 20: "larger degree of precipitation" -> "larger variability in the precipitation": here you want to explain the isotopic variability, not its mean value.*

Changed as suggested:
"… but also due to **larger variability in the** precipitation along these pathways."

l) *p 18 l 17-19: to verify these mechanisms, observations won't be sufficient. Large-eddy simulations would be useful. A sentence to mention this could be added.*

Added as suggested:
**"Furthermore, modelling of the isotopic composition in the MBL with various approaches spanning from simple mixing models to large-eddy simulations could help to understand the measured profiles."**

m) *p 19 l 9: "more important" -> "more variable" (same comment as p 15 l 20)*

Changed as suggested:
"Furthermore, moisture loss during transport, which affects the SWI composition of water vapour, is more **variable** in the extratropics than in subtropical and tropical regions."

**References:**

*Aemisegger, F., Spiegel, J. K., Pfahl, S., Sodemann, H., Eugster, W., and Wernli, H.:* Isotope meteorology of cold front passages: A case study combining observations and modeling: Water isotopes during cold fronts, Geophys. Res. Lett., 42, 5652–5660, https://doi.org/10.1002/2015GL063988, 2015.$

*Jabouille, P., Redelsperger, J. L., and Lafore, J. P.:* Modification of Surface Fluxes by Atmospheric Con-
vection in the TOGA COARE Region, Mon. Weather Rev., 124, 816–837,
https://doi.org/10.1175/1520-
0493(1996)124¡0816:MOSFBA¿2.0.CO;2, 1996.

*Papritz, L., Pfahl, S., Rudeva, I., Simmonds, I., Sodemann, H., and Wernli, H.:* The role of extratropical cyclones and fronts for Southern Ocean freshwater fluxes, Journal of Climate, 27, 6205–6224, https://doi.org/10.1175/JCLI-D-13-00409.1, 2014.

*Salmon, O. E., Welp, L. R., Baldwin, M. E., Hajny, K. D., Stirm, B. H., and Shepson, P. B.:* Vertical profile
observations of water vapor deuterium excess in the lower troposphere, Atmos. Chem. Phys., 19, 11 525–11 543, https://doi.org/https://doi.org/10.5194/acp-19-11525-2019, 2019.

*Sherwood, S. C., Roca, R., Weckwerth, T. M., and Andronova, N. G.:* Tropospheric water vapor, convection, and climate, Rev. Geophys., 48, RG2001, https://doi.org/10.1029/2009RG000301, 2010.

Simmonds, I. and King, J. C.: Global and hemispheric climate variations affecting the Southern Ocean, Antarct. Sci., 16, 401–413, https://doi.org/10.1017/S0954102004002226, 2004.

*Sodemann, H., Aemisegger, F., Pfahl, S., Bitter, M., Corsmeier, U., Feuerle, T., Graf, P., Hankers, R., Hsiao,*

*G., Schulz, H., Wieser, A., and Wernli, H.:* The stable isotopic composition of water vapour above Corsica during the HyMeX SOP1 campaign: insight into vertical mixing processes from lower-tropospheric survey flights, Atmos. Chem. Phys., 17, 6125–6151, https://doi.org/10.5194/acp-17-6125-2017, 2017.

*Tanny, J. and Cohen, J.:* Revisiting the boundary layer structure used in Craig and Gordon's model of isotope fractionation in evaporation, Isotopes Environ. Health. Stud., 44, 11–21, https://doi.org/10.1080/10256010801887091, 2008.

Toffoli, A., Proment, D., Salman, H., Monbaliu, J., Frascoli, F., Dafilis, M., Stramignoni, E., Forza, R., Manfrin, M. and Onorato, M., 2017. Wind generated rogue waves in an annular wave flume. *Physical review letters, 118*(14), p.144503.

*Young, I.:* Wind generated ocean waves, vol. 2, Elsevier, 1999.

---

## Author Comment (AC2) · 19 Jan 2020

Authors' reply to comment on *Meridional and vertical variations of the water vapour isotopic composition in the marine boundary layer over the Atlantic and Southern Ocean* by Anonymous Referee #2

We would like to thank the reviewer for taking the time to read our manuscript and giving detailed feedback, which helped us to improve the manuscript. Please find our answers to the comments (*in italics*) in the following. Citations from the paper are marked in blue, and **blue bold** refers to text added/changed in the revised manuscript.

**1 Major comments**

*1. Uncertainty evaluation: The second message of the manuscript (section 4.2) details the vertical differences between two infrared spectrometers that were installed at 8 and 13 meters, respectively. On average, significant differences are observed between these two instruments. A significant amount of work is dedicated in this manuscript into characterising the instruments performances. Yet, the results in section 4.2 do not include an error bar for which the differences are significant between the two instruments. In particular, in Fig. 10, a significant number of datapoints presented have very small difference (< 0.2 ‰ in δ18O for instance). Considering the precisions of the instruments (in particular the 2120), it is difficult to assess the relevance of these datapoints. This is a key aspect to be able to justify the wind speed dependency, and it seems that most of the results necessary to evaluate the statistical significance of the results are already presented here. I would suggest make use of the standard deviation of the differences (for instance in Fig. 5) and use pertinent statistical tests (for instance Kruskall Wallis tests) to evaluate in which cases are the differences statistically significant.*

Thank you for pointing this out. We included the uncertainty introduced by the post-processing of the SWI measurements in Fig. 10, which shows that the standard deviation of the vertical differences in each bin is larger than the uncertainty due to the post-processing.
Furthermore, we conducted a Kruskall Wallis test for the three wind regimes to test for statistically significant differences in the vertical SWI gradients of the three regimes. The three groups differ significantly for $\delta^{18}O$, $\delta^{2}H$ and $d$ according to the Kruskall Wallis test ($p$-value on the order of $10^{-12}$). As highlighted by Nicholls (2001), the interpretation of null-hypothesis significance tests needs to be done carefully as the sample size strongly influences the outcome of such tests with a higher probability for rejecting the null hypothesis with increasing sample size. Therefore, we prefer to discuss the confidence intervals (instead of the significance test) in this study.

*2. In the manuscript, the authors do not provide any quantitative evaluation of the vertical differences of the isotopic composition in the marine boundary layer. Yet, formulations have been predicted, based on very limited number of observations compared to this study. While I generally agree with the qualitative proposition of the authors, I believe that they should have tested previous formulations. From articles already mentioned in the manuscript, I would suggest to compare their results to models of isotopes in the boundary layers, namely Craig (1965), Merlivat (1978), or again Benetti et al. (2018). I would suggest to use formulations developed in Cappa et al. (2003), and the parametrisations of Merlivat (1978) for the dependency of the diffusion with turbulence. I suggest that these parametrisations, which already include an increasing impact of turbulence with wind speed, should be tested. Due to the considerable amount of data of the authors, I would suggest evaluating this on typical cases (for instance, the regimes [I], [II] and [III] identified by the authors. Also, as here δ18O is expected to decrease monotonously with height, I would suggest that the authors identify the different contributions to d-exc and δ18O (or δD and δ18O) in an isotope-isotope space (for instance*

*δD vs δ18O) and illustrate which process is characterized with slopes higher or lower than the meteoric water line.*

Thank you for this proposition. Our response to this comment addresses three points brought up by the reviewer:

1) **Application of existing models such as Craig and Gordon (1965), Merlivat (1978) or Benetti et al. (2018):**
   As suggested, we tested previous formulations of isotopic models for the marine boundary layer such as for example the combined evaporation-vertical mixing model by Benetti et al. (2018) to predict the SWI composition at 8 m a.s.l from the SWI measurements at 13 m. The SWI-13 measurements provide the isotopic composition of the air parcels mixed in from above and the ocean evaporation flux is defined by the Craig-Gordon model (see Benetti et al. 2018 for more details). However, because we do not have calibrated specific humidity data for SWI-8, we have to estimate the fraction of water vapour that is mixed downward based on an estimated specific humidity profile. Note that due to logistic reasons, it is not possible to calibrate the SWI-8 measurements a posteriori. Our results from applying the Benetti et al. model in this setup shows that the predicted isotope time series for SWI-8 strongly depends on the chosen value of the specific humidity at 8 m a.s.l. Furthermore, we think that the large variety of vertical SWI profiles measured in this study and in more detailed aircraft-based studies such as Sodemann et al. (2017) and Salmon et al. (2019) show that the processes involved in shaping these vertical profiles are likely more complex. In particular, we think that horizontal advection and the formation of convective plumes can lead to a variety of profiles that cannot be predicted without a 3D numerical model (isotope-enabled LES or high resolution regional numerical weather prediction model). We therefore do not find it straightforward to use existing simple models to predict the vertical gradients within the marine boundary layer. Please also note that detailed simulations with the isotope-enabled model COSMOiso have been performed around Antarctica and will be compared to the ACE measurement data in a follow-up study.

2) **The use of the formulations developed in Cappa et al. (2003), and the parametrisations of Merlivat (1978) for the dependency of the diffusion on turbulence:**
   A closer investigation of the dependence of the non-equilibrium fractionation factor on diffusion and turbulence within the three regimes presented in the manuscript based on the ACE data is an excellent idea. We thank the reviewer for this suggestion.

   We used the non-equilibrium fractionation factor as described by Cappa et al. (2003) to calculate the relative importance of turbulent and diffusive transport. In their equation 5, Cappa et al. (2003) described the non-equilibrium fractionation factor as the ratio of the molecular diffusivity of the heavy isotope ($D_H$) and the light isotope ($D_L$) to the power of $n$. Here we use the diffusivity ratios from Merlivat (1978).

$$\alpha_{\text{diffusion}}^{*} \frac{K_H}{K_L} = \left[\frac{D_H}{D_L}\right]^n,$$

*Equation 5 from Cappa et al. (2003).*

The exponent $n$ is equal to zero if the transport during ocean evaporation is completely turbulent (i.e. if no non-equilibrium fractionation occurs), and equal to 1 if the transport is completely diffusive. Therefore, if $n$ increases, non-equilibrium fractionation becomes more important. One way to estimate $n$ from point measurements is to use an existing relation between the $d$-$h_s$ slope, where $h_s$ is the relative humidity with respect to the sea surface temperature, and the exponent $n$ that is based on the linearised Craig Gordon model and the closure assumption (see Aemisegger and Sjolte 2018 for more details). In a purely turbulent regime, in which no non-equilibrium fractionation occurs, $d$ is insensitive to changes in $h_s$ and therefore the $d$-$h_s$ slope is 0. In the case of a purely diffusive regime, $d$ is very sensitive to $h$ and therefore the $d$-$h_s$ slope is steep. The following expression can be found using the linearized Craig Gordon model linking the $d$-$h_s$ slope and the exponent $n$ (see Aemisegger and Sjolte 2018 for more details): $n = -0.53045 * s_{d(hs)} - 0.00699$, where $s_{d(hs)}$ is the slope of the linear relation between $d$ and $h_s$. We calculated $s_{d(hs)}$ for legs 1-3 using the 1-hourly measurement points within 3-day running windows using the measurements at 13 m.

As expected and shown in Fig. A, $n$ increases with decreasing wind speed. During periods with large differences in $\delta^{18}O$ and $d$ between the two measurement heights, high values of $n$ are often observed. The mean $n$ over all legs is 0.128±0.070, which lies below the values (0.22 − 0.25) found by other studies (Gat et al., 1996; Pfahl and Wernli, 2008). The reason for the low $n$ in this study is the flatter $d$-$h_s$ slope of −0.38‰/% (for RHsst<1.0, when ocean evaporation can occur) compared to slopes between −0.57‰/% and −0.42‰/% from previous studies using measurements (Steen-Larsen et al., 2014, 2015; Benetti et al., 2015; Uemura, 2008). A slope of −0.38‰/% however lies within the range of values obtained for the Southern Ocean by Aemisegger and Sjolte (2018) using the closure assumption and the Craig Gordon model based on ERAinterim reanalysis data (compare their Fig. 9b).

[Figure]

*Figure A: Bi-dimensional histograms of n versus wind speed (left), $\Delta_{13-8}\delta^{18}O$ (middle) and $\Delta_{13-8}d$ (right), coloured by number of points per bin. n is calculated using all 1-hourly measurement points of $h_s$ and d within a 3-day moving window and the linearised Craig and Gordon model (Aemisegger and Sjolte 2018). The vertical black line denotes the mean value of n over all legs of 0.128. Wind speed, $\delta^{18}O$ and d are 72-hour moving averages. Values of n<0 are overlaid by white hatches. Note that for these measurement periods $h_s$ was larger than 1.0 indicating dew deposition, in which the above framework for evaporative conditions is not valid.*

**3) Analysis of three regimes in the δD-δ¹⁸O phase space and comparison to the meteoric water line.**

We compared the three wind regimes in terms of their behaviour of the isotope measurements in the $\delta^{18}O$- $\delta^{2}H$-space (Fig. B). Due to the low SST for a large part of the ACE track for legs 1-3, $d$ is lower than 10 for most of the measurement points. Therefore, these points are below the meteoric water line. For regime I, most of the points are below the meteoric water line. For regime II and III, 12% of all points, most of which with high δ-values, are above the meteoric waterline.

[Figure]

*Figure B: Scatterplots of δ¹⁸O versus δ²H, coloured by d, for the three wind regimes using 1-hourly SWI-13 measurements from legs 1-3. The black line shows the meteoric water line ( δ²H = δ¹⁸O*8+10)*

Because this analysis goes beyond the scope of this study, and to keep the manuscript concise as the editor requested when we submitted the manuscript, we decided not to include these results. Furthermore, we think that the qualitative discussion in the paper is adequate given the unfortunately missing vertical profiles of specific humidity. We agree that a study that investigates the factors influencing the exponent *n* in different synoptic situations and on the relative importance of turbulent and diffusive transport near the ocean surface would be a very useful follow-up of the present study. The (open-access) data set provides many opportunities for additional investigations including a comparison study of different MBL isotope models.

Changes to the manuscript:

• We now mention the idea on an evaluation of various MBL mixing models and a closer analysis of the influence of diffusion during ocean evaporation in future studies with the ACE datasets:

   **"Furthermore, modelling of the isotopic composition in the MBL with various approaches spanning from simple mixing models to large-eddy simulations could help to understand the measured profiles."**

**2 Minor comments**

a) *Page 2, Line 10: "The atmospheric water cycle is an essential component of the Earth's climate system" The water cycle is not just atmospheric by definition.*

   Changed to: "The atmospheric **branch of the** water cycle is..."

b) *Page 2, line 24:*
   *"SWIs are tracers of moist atmospheric processes because they record phase changes in the atmosphere." What is a moist atmospheric process ? Sentence unclear*

   Changed to: "…SWIs are tracers of **atmospheric processes involving phase changes of water.**"

c) *Page 3, line 28 to 35: I would suggest include articles such as (Craig, 1965; Cappa et al., 2003).*

   We added two sentences (pages 3/4) on the modelling of SWIs in the MBL during evporative conditions referring to previous studies:

   **"A similar view of the lower MBL, dividing it into a thin laminar layer close to the ocean surface and a turbulent layer above, was used by Craig and Gordon (1965) to calculate the isotopic composition of the evaporative flux from the ocean surface. The Craig-Gordon (1965) model has been applied and refined in various studies and has been shown to adequately simulate the isotopic composition of the MBL water vapour under evaporative conditions (e.g. Merlivat and Jouzel, 1979; Gat, 2008; Horita et al., 2008; Pfahl and Wernli, 2009; Benetti et al., 2018; Feng et al., 2019)."**

d) *Page 4, line 5 to 19: The link with the isotopes and their limits in this context is missing.*

   We added a sentence about why we are interested in turbulence in MBL/close to ocean surface:
   **"Ship-based measurements are normally situated in the surface layer of the MBL and thus directly influenced by turbulent conditions."**

e) We added a sentence addressing the limitation of using isotopes to analyse sea spray evaporation:
   **It is difficult to directly measure sea spray evaporation and, therefore, it is still an open question to what extent sea spray evaporation affects moisture in the MBL (Veron et al., 2015) in different wind forcing conditions.**

**References:**

*Aemisegger, F. and Sjolte, J.:* A climatology of strong large-scale ocean evaporation events. Part II: Relevance for the deuterium excess signature of the evaporation flux, J. Clim., 31, 7313–7336, https://doi.org/10.1175/JCLI-D-17-0592.1, 2018.

*Benetti, M., Aloisi, G., Reverdin, G., Risi, C., and Sèze, G.:* Importance of boundary layer mixing for the isotopic composition of surface vapor over the subtropical North Atlantic Ocean, J. Geophys. Res. Atmos., 120, 2190 – 2209, https://doi.org/10.1002/2014JD021947, 2015.

*Benetti, M., Lacour, J.-L., Sveinbjörnsdóttir, A. E., Aloisi, G., Reverdin, G., Risi, C., Peters, A. J., and Steen-Larsen, H. C.:* A framework to study mixing processes in the marine boundary layer using water vapor isotope measurements, Geophys. Res. Lett., 45, 2524–2532, https://doi.org/10.1002/2018GL077167, 2018.

*Cappa, C. D., Hendricks, M. B., DePaolo, D. J., and Cohen, R. C.:* Isotopic fractionation of water during evaporation, J. Geophys. Res. Atmos., 108, 4525, https://doi.org/10.1029/2003JD003597, 2003.

*Craig, H. and Gordon, L.:* Deuterium and oxygen 18 variations in the ocean and the marine atmosphere, in: Proceedings of the Stable Isotopes in Oceanographic Studies and Paleotemperatures, 1965.

*Dütsch, M., Pfahl, S., and Sodemann, H.:* The impact of nonequilibrium and equilibrium fractionation on two different deuterium excess definitions, J. Geophys. Res.: Atmos., 122, 12732–12746, https://doi.org/10.1002/2017JD027085, 2017.

*Feng, X., Posmentier, E. S., Sonder, L. J., and Fan, N.:* Rethinking Craig and Gordon's approach to modeling isotopic compositions of marine boundary layer vapor, Atmos. Chem. Phys., 19, 4005–4024, https://doi.org/10.5194/acp-19-4005-2019, 2019.

*Gat, J. R.:* Oxygen and hydrogen isotopes in the hydrologic cycle, Annu. Rev. Earth Planet. Sci., 24, 225–262, https://doi.org/10.1146/annurev.earth.24.1.225, 1996.

*Gat, J. R.:* The isotopic composition of evaporating waters – review of the historical evolution leading up to the Craig–Gordon model, Isotopes in Environmental and Health Studies, 44, 5–9, https://doi.org/10.1080/10256010801887067, 2008.

*Horita, J., Rozanski, K., and Cohen, S.:* Isotope effects in the evaporation of water: a status report of the Craig–Gordon model, Isotopes Environ. Health. Stud., 44, 23–49, https://doi.org/10.1080/10256010801887174, 2008.

*Merlivat, L.:* Molecular diffusivities of $H_2{}^{16}O$, $HD^{16}O$, and $H_2{}^{18}O$ in gases, The Journal of Chemical Physics, 69, 2864, https://doi.org/10.1063/1.436884, 1978.

*Merlivat, L. and Jouzel, J.:* Global climatic interpretation of the deuterium-oxygen 18 relationship for precipitation, J. Geophys. Res. Oceans, 84, 5029–5033, https://doi.org/10.1029/JC084iC08p05029, 1979.

Nicholls, N.: "The insignificance of significance testing", Bulletin of the American Meteorological Society, Vol. 82 No. 5, pp. 981-986, 2001.

Pfahl, S. and Wernli, H.: Lagrangian simulations of stable isotopes in water vapor: An evaluation of nonequilibrium fractionation in the Craig-Gordon model, J. Geophys. Res.: Atmos., 114, D20108, https://doi.org/10.1029/2009JD012054, 2009.

Salmon, O. E., Welp, L. R., Baldwin, M. E., Hajny, K. D., Stirm, B. H., and Shepson, P. B.: Vertical profile observations of water vapor deuterium excess in the lower troposphere, Atmos. Chem. Phys., 19, 11525–11 543, https://doi.org/https://doi.org/10.5194/acp-19-11525-2019, 2019.

Samuels-Crow, K. E., Galewsky, J., Sharp, Z. D., and Dennis, K. J.: Deuterium excess in subtropical free troposphere water vapor: Continuous measurements from the Chajnantor Plateau, northern Chile, Geophys. Res. Lett., 41, 8652–8659, https://doi.org/10.1002/2014GL062302, 2014.

Sodemann, H., Aemisegger, F., Pfahl, S., Bitter, M., Corsmeier, U., Feuerle, T., Graf, P., Hankers, R., Hsiao, G., Schulz, H., Wieser, A., and Wernli, H.: The stable isotopic composition of water vapour above Corsica during the HyMeX SOP1 campaign: insight into vertical mixing processes from lower-tropospheric survey flights, Atmos. Chem. Phys., 17, 6125–6151, https://doi.org/10.5194/acp-17-6125-2017, 2017.

Steen-Larsen, H. C., Sveinbjörnsdottir, A. E., Peters, A. J., Masson-Delmotte, V., Guishard, M. P., Hsiao, G., Jouzel, J., Noone, D., Warren, J. K., and White, J. W. C.: Climatic controls on water vapor deuterium excess in the marine boundary layer of the North Atlantic based on 500 days of in situ, continuous measurements, Atmos. Chem. Phys., 14, 7741–7756, https://doi.org/10.5194/acp-14-7741-2014, 2, 2014.

Steen-Larsen, H. C., Sveinbjörnsdottir, A. E., Jonsson, T., Ritter, F., Bonne, J.-L., Masson-Delmotte, V., Sodemann, H., Blunier, T., Dahl-Jensen, D., and Vinther, B. M.: Moisture sources and synoptic to seasonal variability of North Atlantic water vapor isotopic composition, J. Geophys. Res. Atmos., 120, 2015JD023 234, https://doi.org/10.1002/2015JD023234, 2015.

Uemura, R., Matsui, Y., Yoshimura, K., Motoyama, H., and Yoshida, N.: Evidence of deuterium excess in water vapor as an indicator of ocean surface conditions, J. Geophys. Res., 113, D19114, https://doi.org/10.1029/2008JD010209, 2008.

Veron, F.: Ocean spray, Annu. Rev. Fluid Mech., 47, 507–538, https://doi.org/10.1146/annurev-fluid-010814-014651, 2015.

---

## Author Comment (AC3) · 19 Jan 2020

Authors' reply to comment on *Meridional and vertical variations of the water vapour isotopic composition in the marine boundary layer over the Atlantic and Southern Ocean* by Anonymous Referee #3

We would like to thank the reviewer for taking the time to read our manuscript and giving detailed feedback, which helped us to improve the manuscript. Please find our answers to the comments (*in italics*) in the following. Citations from the paper are marked in blue, and **blue bold** refers to text added/changed in the revised manuscript.

**1 Technical corrections**

a) *Page 2, line 24: Unless you think that this is obvious, I recommend the authors add that R(VSMOW2) is multiplied by 2 for the two possible positions of the isotope within the water molecule.*

Thank you for pointing this out. We added the proposed information:

"… (VSMOW2; with $^2RVSMOW2 = 1.5576 \cdot 10^{-4}$ and $^{18}RVSMOW2 = 2.0052 \cdot 10^{-3}$ ; **$^2$RVSMOW2 is multiplied by two due to the two possible positions of $^2$H within the water molecule**)."

b) *Page 2, line 26: Equilibrium fractionation is not the only type of isotopic fractionation. Non-equilibrium fractionation by diffusion is also isotopic fractionation. I recommend the authors modify as follows: 'The difference in saturation vapor pressure between heavy and light isotopologues causes one type of isotopic fractionation . . .'*

Changed to:
"The difference in saturation vapour pressure between heavy and light isotopes causes **one type of isotopic fractionation, ...**"

c) *Page 9, line 14: The authors state that horizontal differences between SWI-8-sb ad SWI-8-ps sensors are smaller than vertical differences, but Figure 5 seems to indicate that d has large horizontal differences comparable to vertical differences in d.*

Yes, the horizontal SWI differences can be as large as the vertical differences for short periods, but during most of the time, the horizontal differences are smaller than the vertical differences. The mean horizontal differences (with 65% confidence range) are 0.8 [-1.6...3.2]‰ for $\delta 2H$, -0.04 [-0.41...0.38]‰ for $\delta 18O$ and 1.2[-0.2...2.4] ‰ for *d* (see also Appendix Fig. A1) compared to -2.6 [-4.8...-0.2]‰ for $\delta 2H$, -0.55 [-0.90...-0.14]‰ for $\delta 18O$  and 1.8 [0.5...3.2] ‰ for *d* for the vertical differences.

d) *Page 13, line 23 incorrect figure citation, should be: 'The meridional distribution of d (Fig. 6c) . . .'*

Changed as suggested

e) *Page 14, line 17 incorrect figure citation, should be: Fig. 6 a,b,c.*

Changed as suggested

f) *Page 15, line 25 incorrect figure citation, should be: Fig. 6f.*

Changed to Fig. 6c,f to refer to both, $d_{IQR}$ and $h_{SST,IQR}$

g) *Figure 1, page 26: I suggest that the authors add latitude labels on Figure 1, it would help with interpretation.*

Changed as suggested

**2 Spelling/Typos**

a) *Page 6, line 12 (also lines 19, 20, 33): Is apostrophe 12'000 standard Swiss notation for 12 000? I recommend no apostrophe to avoid confusion.*

Changed as suggested

b) *Page 13, line 6: change 'warmer T' to 'higher T'.*

Changed as suggested

c) *Page 13, line 12: spelling typo, should be 'meridional'.*

Changed as suggested

d) *Page 13, line 33: spelling typo, should be 'Agulhas'.*

Changed as suggested

e) *Figure 4, page 29: add abscissa (x-axis) label: "Date (dd-mm)".*

Changed as suggested

f) *Figure 5, page 30: add abscissa (x-axis) label: "Date (dd-mm)".*

Changed as suggested

g) *Figure 6, page 31: the letters 6a, 6b, etc. in the text do not appear to match the order of figures in Figures 6. Please recheck.*

Checked and changed where needed.

h) *Figure 6 caption, page 31, and Page 18, line 32: I recommend changing 'site' to 'location' because site implies a fixed location whereas you are measuring at many latitudes along the ship track.*

Changed as suggested

i) *Figure 10 caption, page 35: in the next-to-last sentence change 'Less points' to 'Fewer points. . .'*

Changed as suggested